# Structural Diversity, Characterization and Toxicology of Microcystins

**DOI:** 10.3390/toxins11120714

**Published:** 2019-12-07

**Authors:** Noureddine Bouaïcha, Christopher O. Miles, Daniel G. Beach, Zineb Labidi, Amina Djabri, Naila Yasmine Benayache, Tri Nguyen-Quang

**Affiliations:** 1Écologie, Systématique et Évolution, Univ. Paris-Sud, CNRS, AgroParisTech, Université Paris-Saclay, 91405 Orsay, France; djabri-amina@univ-eltarf.dz (A.D.); nailayasmine.benayache@umc.edu.dz (N.Y.B.); 2Biotoxin Metrology, National Research Council Canada, 1411 Oxford St, Halifax, NS B3H 3Z1, Canada; christopher.miles@nrc-cnrc.gc.ca (C.O.M.); daniel.beach@nrc-cnrc.gc.ca (D.G.B.); 3Laboratoire Biodiversité et Pollution des Écosystèmes, Faculté des Sciences de la Nature et de la Vie, Université Chadli Bendjedid d’El Taref, 36000 El Taref, Algeria; labozina23@gmail.com; 4Biofluids and Biosystems Modeling (BBML), Faculty of Agriculture, Dalhousie University, 39 Cox Road, Truro, B2N 5E3 Nova Scotia, Canada; Tri.nguyen-quang@dal.ca

**Keywords:** microcystin, cyanobacteria, cyanotoxin, structural elucidation, toxicology

## Abstract

Hepatotoxic microcystins (MCs) are the most widespread class of cyanotoxins and the one that has most often been implicated in cyanobacterial toxicosis. One of the main challenges in studying and monitoring MCs is the great structural diversity within the class. The full chemical structure of the first MC was elucidated in the early 1980s and since then, the number of reported structural analogues has grown steadily and continues to do so, thanks largely to advances in analytical methodology. The structures of some of these analogues have been definitively elucidated after chemical isolation using a combination of techniques including nuclear magnetic resonance, amino acid analysis, and tandem mass spectrometry (MS/MS). Others have only been tentatively identified using liquid chromatography-MS/MS without chemical isolation. An understanding of the structural diversity of MCs, the genetic and environmental controls for this diversity and the impact of structure on toxicity are all essential to the ongoing study of MCs across several scientific disciplines. However, because of the diversity of MCs and the range of approaches that have been taken for characterizing them, comprehensive information on the state of knowledge in each of these areas can be challenging to gather. We have conducted an in-depth review of the literature surrounding the identification and toxicity of known MCs and present here a concise review of these topics. At present, at least 279 MCs have been reported and are tabulated here. Among these, about 20% (55 of 279) appear to be the result of chemical or biochemical transformations of MCs that can occur in the environment or during sample handling and extraction of cyanobacteria, including oxidation products, methyl esters, or post-biosynthetic metabolites. The toxicity of many MCs has also been studied using a range of different approaches and a great deal of variability can be observed between reported toxicities, even for the same congener. This review will help clarify the current state of knowledge on the structural diversity of MCs as a class and the impacts of structure on toxicity, as well as to identify gaps in knowledge that should be addressed in future research.

## 1. Introduction

There are an increasing number of warnings about toxic cyanobacterial blooms observed worldwide and global warming is thought to stimulate their development in eutrophic waters [1,2,3,4,5]. These blooms are often accompanied by production of a variety of cyanotoxins generally classified according to the target organs: hepatotoxins (liver), neurotoxins (nervous system), and dermatotoxins (skin). Among these cyanotoxins, it appears that hepatotoxic and tumor promoting microcystins (MCs) are more commonly found in cyanobacterial blooms and considered to be one of the most hazardous groups throughout the world [6,7,8,9,10]. Despite the significant amount of available information, interest in MCs continues to increase due to their well-known hazards to farm animals, fisheries, aquaculture, human health, and wildlife through exposure via drinking, environmental, and recreational waters [7,8,10,11,12].

Louw and Smit [13] were among the first scientists who isolated and attempted to characterize a cyanotoxin from a cyanobacterial bloom, dominated by *Microcystis toxica*, which occurred in the Vaal Dam reservoir in South Africa in 1942–1943. They concluded that it was an alkaloid of undetermined structure with acute and chronic hepatotoxic properties. Initial attempts to characterize the structures of MCs started in the late 1950s [14]. However, full structural identification of the first MC congeners was achieved in 1984, when a combination of amino acid analysis, nuclear magnetic resonance (NMR) spectroscopy, and mass spectrometry (MS) were used [15,16]. These and subsequent studies showed the chemical structure of these toxins to consist of a cyclic heptapeptide composed of five relatively conserved amino acids plus two variable L-amino acids [14,15,16,17,18,19,20].

Two decades later, studies have shown that these toxins are biosynthesized nonribosomally via an MC synthetase gene cluster (*mcy*), consisting of a combination of polyketide synthases (PKSs), nonribosomal peptide synthetases (NRPSs) and tailoring enzymes [21,22,23,24]. Recent studies have reported the bulk of this structural diversity is the result of genetic and/or environmental factors, which have an impact on the functioning of enzymes encoded in the *mcy* gene cluster [25,26,27,28,29,30,31,32]. This *mcy* gene cluster is often spontaneously modified through point mutations, deletions and insertions, or a series of genetic recombinations, which affect the functioning of the MC peptide synthetases and result in the chemical diversity observed in nature [33,34,35,36,37].

As a result of the large number of literature reports spanning over four decades of research, it can be difficult to glean accurate information on the total number of identified MCs. The phrase ‘more than 100 microcystin congeners’ is still often used in the literature [12,38,39,40,41] however estimates as high as 248 known MCs have recently been published [42]. In this review, we update this number to 279 and describe the methods by which this structural elucidation was carried out. In addition, we provide an in-depth review of their toxic potential and a discussion of the structure–activity relationships this information provides.

## 2. Nomenclature and General Chemical Structure of Microcystins

Hughes et al. [43] first described a hepatotoxic *‘*fast death factor’ in an extract of the isolated strain *Microcystis aeruginosa* NRC-1. This hepatotoxic factor was later renamed microcystin, derived from the genus *Microcystis* [44,45,46]. Since, they have also been referred to in the literature as cyanoginosin, with prefix ‘cyano’ from the term cyanobacteria and ‘ginosin’ derived from *aeruginosa* [19,47]; and cyanoviridin, with the root ‘viridin’ from the species *M. viridis* [48].

After almost two decades of structural analysis of toxic peptides from the colonial bloom-forming cyanobacterium *M. aeruginosa*, Botes et al. [15,16,17,18,19] and Santikarn et al. [20] provided structural details of these toxins with the general structure of: cyclo-(D-Ala^1^-X^2^-D-Masp^3^-Z^4^-Adda^5^-D-γ-Glu^6^-Mdha^7^) (Figure 1) in which X and Z are variable L-amino acids, D-Masp^3^ is D*-erythro-*β*-*methyl-isoaspartic acid, and Mdha is *N*-methyldehydroalanine. The β-amino acid Adda, (2*S*,3*S*,4*E*,6*E*,8*S*,9*S*)-3-amino-9-methoxy-2,6,8-trimethyl-10-phenyldeca-4,6-dienoic acid, is the most unusual substructure, and has not been reported elsewhere in nature apart from in the closely related group of toxic pentapeptides, the nodularins. After a time, leading researchers in the field warned that the continued use of multiple naming systems could generate confusion as the number of publications on these cyclic peptides increased [49]. Therefore, a universal system of nomenclature was proposed based on the original term microcystin-XZ, where X and Z denote the variable amino acid residues in positions 2 and 4 [50].

The number of fully and partially characterized MCs increased significantly throughout the 1990s and structural variations have been reported in all seven amino acids, but most frequently with substitution of the two variable L-amino acids X and Z at positions 2 and 4, respectively, and demethylation of amino acids at position 3 and/or 7. Multiple combinations of the variable L-amino acids (X and Z), and modifications to the other amino acids, lead to the high levels of structural diversity observed in MCs which have been characterized to date from bloom samples and isolated strains of *Microcystis*, *Oscillatoria*, *Planktothrix*, *Anabaena* (syn. *Dolichospermum*), and *Nostoc* spp. [51]. While Carmichael et al. [50] originally proposed abbreviation in the format MCYST-XZ, in recent decades the abbreviation MC-XZ has come in to general use, where X and Z are the one-letter amino acid abbreviations (where these exist) and with any variations at positions 1, 3, and 5–7 (relative to D-Ala, D-Masp, Adda, D-Glu, and Mdha, respectively) shown in square brackets, using 3–7-letter amino acid abbreviations [52], in numerical order with the position indicated by a superscript, and separated by commas without spaces in square brackets immediately prior to MC. For example, [D-Leu^1^,D-Asp^3^,Dha^7^]MC-LR contains D-Leu, D-Asp and Dha at positions 1, 3, and 7, and L-Leu and L-Arg at positions 2 and 4, respectively, with Adda and D-Glu assumed by default at positions 5 and 6 (see Figure 1). If an amino acid residue at positions 2 or 4 is not one of the 20 standard amino acids, the three-letter (or more, where necessary) abbreviation is used; e.g., the congener containing Leu at position 2 and homoarginine in position 4 is named MC-LHar. Ring opened MCs are designated with the prefix [seco-a/b], where a and b are the residue numbers between which the amide bond has been hydrolysed. For example, [seco-4/5]MC-LR indicates MC-LR hydrolysed between the Arg^4^- and Adda^5^-residues. A MC name-generator is included in version 16 of the MC mass calculator tab of a publicly available toxin mass list [53]. To facilitate ongoing efforts to maintain comprehensive lists and databases of toxins and/or cyanobacterial metabolites, it is recommended that researchers use these naming conventions going forward when reporting the identification of new MCs.

## 3. Biosynthesis of Microcystins

The nonribosomal biosynthesis of MCs has been described in detail in *Microcystis* [21,22,23,54], *Planktothrix* [55], and *Anabaena* [56]. This MC synthetase system is encoded by two transcribed operons in *M. aeruginosa*, *mcyABC*, and *mcyDEFGHIJ*, from a central bidirectional promoter located between *mcyA* and *mcyD* [23,54]. Insertional gene knockout experiments have demonstrated that all MC congeners produced by a strain are synthesized by a single enzyme complex encoded by a 55 kb gene cluster [21,55,57,58]. Therefore, strains that do not produce MCs can result from inactivation either by point mutation, or by partial or complete deletion of *mcy* genes [36,57,59,60]. Recently, Shishido et al. [61] reported that the *mcy* gene cluster did not encode all enzymes for the synthesis of rare variants of MC that contain a selection of homo-amino acids by the benthic cyanobacterium *Phormidium* sp. LP904c, but may instead activate homo-amino acids produced during the synthesis of anabaenopeptins.

Several studies have assessed the impact of environmental factors, including nutrient availability, light, iron limitation, temperature, and pH on cellular MC content [25,31,32,62,63,64,65,66,67,68]. Some of these factors are involved in enhancing or suppressing the expression of the *mcy* gene, leading to an increase or decrease in MC production. For example, Kaebernick et al. [62] reported that light quality has a significant effect on the transcription levels of the genes *mcyB* and *mcyD*. Recently, Yang and Kong [69] reported that the exposure of *M. aeruginosa* cells to UV-B not only remarkably inhibited the growth of *Microcystis*, but also led to a reduction in MC concentration by decreasing transcription of the gene *mcyD*. Sevilla et al. [64] demonstrated that iron starvation causes an increase in *mcyD* transcription and MC levels. Moreover, it was also shown that stress conditions caused by nutrient deprivation increased *mcyD* transcription and MC production [68]. Horst et al. [70] combined laboratory and field experiments and observed a correlation between nitrogen limitation and lower MC cell quotas in a field survey of Lake Erie (situated at the international boundary between Canada and the USA). Since MC is a nitrogen-rich metabolite, Van de Waal et al. [30] have observed an increase in MC cell quotas after excess nitrogen supply, the nitrogen-rich variant MC-RR being the most significantly affected. Similarly, Puddick et al. [32] recently reported that when *Microcystis* strain CAWBG11 was grown in batch culture until nitrogen-depletion, the relative abundance of the MC congeners shifted and the relative abundance of arginine-containing MCs decreased, whilst the congeners which did not contain arginine increased. These changes coincided with a large decrease in nitrate concentration to 0.35% of initial levels.

Therefore, the biosynthesis of MCs is strain dependent and their diversity is due to variability in the coding of the MC synthetase genes among cyanobacterial strains [71] as well as some environmental factors such as nitrogen concentration [29,30,32,72,73]. The large variety of amino acids that can be incorporated and further modified by tailoring enzymes, having activities such as the epimerization, cyclisation, *N*-methylation, formylation, and reduction of amino acids, accounts for the production of highly complex peptides [74,75]. In the field, it is important to consider that multiple strains of cyanobacteria co-exist, often leading to more complex profiles of MCs. Typically, cyanobacterial strains in the field and in controlled conditions produce up to twenty MC congeners but usually only one or two congeners dominate in any single strain [12,40,76,77,78,79,80]. However, the number of congeners produced in a single strain can exceed 47 when all minor MC analogues are considered [33] but in many other cases MC profiles reportedly contain only a small number of congeners. This highlights the fact that there are still new MCs being constantly identified, with varying degrees of certainty, in the very recent literature [81,82]. However, gaps remain in the understanding of how environmental factors can modulate the relative abundance of MC congeners in a bloom, requiring further research on the topic.

## 4. Structural Elucidation of Microcystins

Definitive structural elucidation of new MCs can only be achieved by purifying individual MCs using preparative isolation techniques followed by comprehensive NMR, amino acid analysis and tandem MS (MS/MS) experiments.

Proper purification, structural elucidation, and quantitation are also of critical importance to toxicity studies, where the knowledge of structure, purity, and amount of each MC tested for toxicity are required to ensure the validity and comparability of toxicology results. In this section we give a brief overview of the techniques available for structural elucidation of MCs and the structural information that each technique can provide. This will be useful in choosing appropriate methods for structural elucidation as well as in the critical evaluation of published MC identifications and toxicities.

### 4.1. Mass Spectrometry for Structural Elucidation

In addition to offering sensitive and selective targeted quantitative analysis of known MCs, MS is an invaluable tool for tentative structural elucidation of new MCs, providing information about their molecular formula (full scan) and structure (MS/MS). The types of MS techniques that have been used for the identification of MCs have evolved as mass spectrometers suitable for intact analysis of peptides have become commercially available. Early work relied heavily on direct analysis of MCs purified off-line, using fast atom bombardment MS in a variety of low and high resolution as well as tandem MS configurations [15,75,83,84,85,86]. Similarly, the usually high-resolution MS information from matrix assisted laser desorption ionization-time of flight MS with and without post-source decay has been invaluable in identifying new purified MCs over the years [87,88,89,90,91]. The widespread availability of electrospray ionization (ESI) has made it commonplace to couple analytical liquid chromatography (LC) separations directly to high resolution and/or tandem mass spectrometers. This has offered the possibility for structural characterization and tentative identification of many MCs in complex mixtures without the need for isolation of milligram amounts of toxin needed for full structural elucidation. LC with sequential multi-stage MS dissociation (LC-MS^n^), while usually offering only unit mass resolution, provides a great deal of information for the determination of structure and connectivity and has been widely used for MC identification [33,75,77,90,92,93]. Alternatively, LC with high resolution MS/MS (LC-HRMS/MS) provides accurate mass data on precursor and product ions suitable for determining their molecular formulae [81,82,94,95,96,97].

LC-MS alone is usually not considered definitive proof of chemical structure, but a great deal of structural information can nevertheless be obtained. LC separation of MCs is typically carried out in reverse phase using C_18_ or similar stationary phases and acidic mobile phases, resulting in protonation of carboxyl groups, with acetonitrile as the organic modifier. The hydrophobicity and therefore elution order of MCs in reverse-phase LC is governed primarily by the number of arginine residues in their structure, and the relative elution order of new congeners can provide useful information about structure. MCs with two arginine or homoarginines (e.g., MC-RR) are the most polar and elute first, followed by MCs with a single Arg at position four (e.g., MC-LR), then those with a single Arg at position 2 (e.g., MC-RY), and finally those with no Arg residues (e.g., MC-LA). Though sometimes difficult to separate chromatographically [98], the relative elution order of desmethylated MCs can also be useful for identitication of minor analogues with [DMAdda^5^]-variants eluting much earlier than [D-Asp^3^]-variants, which are followed closely by [Dha^7^]-variants [82].

The nominal molecular weights for known parent MCs range from 882 Da to 1101 Da, although the theoretical range using the known amino acids identified to date in natural MCs is significantly wider than this [53]. These masses are too large for definitive molecular formula elucidation by HRMS when all possible elements are considered (e.g., F, P, Cl, etc.), but when the possible elemental composition and ring or double bond equivalents (RDBE) are limited to a reasonable range similar to known MCs, in most cases a candidate formula can be determined. Careful evaluation of isotopic patterns in HRMS can further help narrow down possible molecular formulae, relying less on pre-determined limits in elements, as reported recently in Mallia et al. [99]. Known parent (unconjugated) MCs have between 44 and 57 C atoms, 63 and 84 H atoms, 7 and 13 N atoms, 12 and 17 O atoms, and up to one S atom in methionine-containing MCs [53]. It can also be useful to limit possible molecular formulas by the number of RDBEs in the structure, which ranges from 17 to 27. Within these limits there are currently no cases among known MC where [M+H]^+^ or [M+2H]^2+^ ions are within 5 ppm mass accuracy from one another. However, there are >120 cases reported of isomeric MCs reported, significantly limiting the utility of accurate mass measurements alone for MC identification. It should be noted that conjugates between MCs and thiol-containing biomolecules (e.g., cysteine, glutathione) have been reported [81,100,101] and that mass ranges and molecular formula tolerances need to be widened accordingly when characterizing these compounds.

The ionization and MS/MS fragmentation of MCs are also generally similar between MCs with the same number of Arg/Har residues. In positive ESI mode, MCs with two Arg/Har residues, or conjugated to Cys or GSH, appear primarily as doubly charged ions after ESI while those with one or no Arg/Har are predominantly singly charged. MCs with no Arg/Har ionize comparatively poorly and therefore tend to form adducts with other cations present in the sample (e.g., [M+Na]^+^ or [M+NH_4_]^+^). In negative ESI all MCs are observed predominantly as singly charged [M-H]^-^ ions. The MS/MS dissociation of MCs has been studied extensively in both positive [102,103], and negative [104,105] ionization modes. In most cases, MCs fragment to produce sensitive class-characteristic fragments at *m*/*z* 135.0804 originating from the Adda^5^–residue in positive mode, and *m*/*z* 128.0353 originating from the Glu^6^-residue in negative mode [82,105,106]. Since MCs are cyclic peptides, a large body of previous work on the collision induced dissociation (CID) of non-microcystin peptides also exists [107] that can be useful in interpretation of microcystin MS/MS spectra. In addition to the class-characteristic fragments, various “sequence ions” are formed during CID following ring opening that represent cleavages of individual amino acids. These can be used to effectively reconstruct the MC structure one amino acid at a time. For example, in negative mode, the B-ion series is useful for determining the mass of the amino acid at position 4 since ions represent successive cleavages from the B1 ion [C_5_H_4_O_3_-X^4^-Adda-(Glu-H_2_O)-Mdha-Ala]^−^ [105]. Similarly, in positive mode, fragments associated with each individual amino acid can be readily identified from HRMS/MS data. This allows the amino acid at which small structural variations, such as demethylation have occurred, to be determined but not always the definitive structure of amino acid side chains, particularly for low resolution MS data [102].

There are two different types of CID that provide different and often complementary information for structural elucidation; quadrupole and ion trap CID. Quadrupole CID, sometimes termed higher energy collisional dissociation (HCD), is a single-stage process involving a smaller number of more energetic collisions. The main advantage of this approach is that the full *m*/*z* range of product ions can be detected simultaneously, allowing for sensitive detection of both class-specific and structurally informative product ions in a single experiment. However, it can be more difficult to trace the origin of lower mass product ions. Ion trap CID involves longer time scales and a larger number of lower energy collisions, as well as offering the potential for multiple stages of MS/MS (MS^n^). Here, structural elucidation is done by constructing a fragmentation tree where the origin of lower mass product ions can be established in higher order experiments (e.g., MS^3^ or MS^4^) in relation to higher-mass product ions observed in MS^2^. The principal limitation of ion trap dissociation is the 1/3 rule, stating that product ions of less than 1/3 the *m*/*z* of a selected precursor ion cannot be isolated for detection. This means that class-specific low mass product ions cannot be detected in MS^2^ in an ion trap regardless of the collision energy used.

Use of established chemical degradation or modification reactions to probe the structure of unknown chemicals is a classical approach that pre-dates the availability of modern analytical instrumentation such as NMR or MS. This approach was an important part of the first structural elucidation studies on MCs, where partial hydrolysis reactions followed by Edman degradation reactions and mass spectrometric methods of characterizing the linearized products were utilised [15,16]. However, when combined with modern LC-MS, similar approaches are even more powerful due to the ease with which multiple products of reactions with multiple reactants can be selectively detected. The reactivity of the double bond in Dha^7^ and Mdha^7^ of MCs towards thiols has emerged as being particularly useful for characterizing MCs congeners [81,82,99]. In particular, mercaptoethanol derivatization has been exploited in such a manner both to identify new MCs from complex samples [77,108] or to quickly differentiate Mdha^7^- from Dhb^7^-containing MCs [109], which is not possible using LC-MS alone unless authentic standards are available. Another useful reaction is methyl esterification that, together with LC-MS, can be used to count the number of reactive carboxylic acid (the CO_2_H group of Glu^6^ is reactive, but not that of Masp^3^/Asp^3^) and phenolic groups in MCs [96,99]. Selective oxidation with mild oxidants can also be combined with LC-MS to identify MCs containing sulfide and sulfoxides, such as methionine (e.g., MC-MR) [81,93]. This approach has also proved useful in helping to identify sulfide (e.g., Cys, GSH) conjugates of MCs, and their autoxidation products, in natural samples [81]. These conjugates are also readily semi-synthesised to produce authentic standards for confirmation of their presence in samples and cultures by LC-MS (e.g., [81,100]) (see Section 5.7). In addition, the identity of *S*-conjugates at position-7 of MCs can be confirmed through a range of deconjugation reactions [101,110,111].

It should be noted that stereochemical information is not generally accessible from LC-MS/MS analyses. It is therefore generally assumed in the literature that the stereochemistry of the amino acids in MCs whose structures are established using LC-MS methods are the same as those in well-established structures such as MC-LR unless there is evidence to the contrary, and the same assumption is made throughout this review.

### 4.2. Preparative Isolation of Microsystins

While a great deal of structural information is available from LC-MS without the need for additional purification of MCs, stereochemistry is not accessible by LC-MS. Definitive structural elucidation by NMR currently requires isolation of 50–100 μg or more of a pure MC. This is most often done by scaling up production of laboratory cultured cyanobacteria, or directly from naturally occurring bloom samples. Dried cyanobacterial biomass is usually first extracted under acidic conditions. Aqueous extracts can then be applied to a reverse phase (e.g., C_18_) open column or large solid phase extraction (SPE) cartridge, depending on the complexity of the toxin profile. Elution is then carried out in a stepwise fashion, usually with aqueous methanol. In some cases, size exclusion chromatography has been used [83] but is often not required. For more complex toxin mixtures, a final semi-preparative isolation step is required, again using a reverse phase column. Additional selectivity at each step can be achieved by varying the pH of the eluent with various buffers. These then need to be removed with an additional SPE step. Long-term storage of MCs in acidic methanol should be avoided because of the potential for formation of methyl esters [81,112] (see Section 5.7).

### 4.3. Amino Acid Analysis

One of the classical approaches to structural characterization of MCs after preparative isolation is to hydrolytically digest them to their constituent amino acids, which are then identified chromatographically based on retention time comparison to amino acid standards. Typically, hydrolysis is done in 6 M HCl at 105 °C for 24 h. It should be noted that under these conditions, Mdha and Adda were not detected [18] and some amino acids (e.g., tryptophan, glutamine, serine, threonine, tyrosine, methionine, and cysteine) are altered [113] and their presence must be inferred. Amino acids can then be derivatized with a UV absorbent tag and analyzed by reverse phase LC-UV [83]. Alternatively, they can be derivatized with trifluoroacetic anhydride to make them volatile and then analyzed by gas chromatography [114]. When chiral stationary phases or derivatizing agents are used, it is possible to determine the absolute configuration of the amino acids, which is not possible by MS or NMR alone.

### 4.4. Structural Elucidation by Nuclear Magnetic Resonance Spectroscopy

Although it is most often used in combination with other approaches mentioned earlier, NMR is usually required for definitive structural identification of MCs including relative stereochemistry. Often CD_3_OD and sometimes D_2_O were used in the past as solvents for NMR spectroscopy of MCs to reduce ^1^H signals arising from the solvent signals in 1- and 2-D spectra e.g., [115]. The disadvantage is that exchangeable amide protons, and thus their correlations, cannot be detected. It is therefore advantageous to use solvents that do not exchange the amide protons, such as (CD_3_)_2_SO or CD_3_OH, and with modern NMR spectrometers and solvent suppression techniques, the use of partially protonated solvents such as CD_3_OH is not usually problematic. However, NMR spectroscopy is a relatively insensitive technique, and significant amounts of purified compound are generally required. With modern high-field NMR spectrometers, full structural elucidations can be performed on less than 50 μg of an MC in CD_3_OH in 5 mm NMR tubes.

A comprehensive approach to assigning all protons and carbons in MCs by NMR alone was presented nearly 3 decades ago that relied on a combination of three 2-D NMR techniques: double-quantum filtered correlation spectroscopy (DQF-COSY), ^l^H-detected multiple quantum coherence (HMQC), and heteronuclear multiple bond correlation (HMBC) NMR spectroscopy [115]. DQF-COSY was first used to assign H-2 of each constituent amino acid along with several other proton assignments. HMQC was then used to assign C-2 of each constituent amino acid based on the H-2 resonances already assigned. Finally, HMBC NMR spectra, that can detect ^2^*J*- and ^3^*J*-couplings between carbon and hydrogen atoms, were used to assign the carbonyl carbon from each amino acid and ultimately to determine the overall amino acid sequence. Nowadays, total correlation spectroscopy (TOCSY) or decoupling in the presence of scalar interactions (DIPSI-2) NMR spectra would typically also be acquired to identify the individual amino acid spin systems, and edited heteronuclear single quantum correlation (HSQC) spectra to identify methyl, methylene, and methine ^13^C and ^1^H resonances. Rotating frame nuclear overhauser effect spectroscopy (ROESY) NMR spectra are generally better suited for detection of through-space correlations for molecules of the size of MCs than are nuclear overhauser effect spectroscopy (NOESY) NMR spectra. In addition to detecting through-space correlations and thus potentially defining the relative stereochemistry of the molecule, ROESY NMR correlations can be useful for establishing the connectivity of the amino acids as a supplement to HMBC NMR correlations. Selective 1D-TOCSY or –DIPSI-2 NMR spectra can often be used to obtain ^1^H signal multiplicities and coupling constants in regions of spectral overlap, which can be useful for chemical shift and stereochemical assignments. The higher resolution afforded in the ^13^C axis by band-selective HMBC and HSQC spectra is sometimes useful for resolving crowded regions of the HSQC (e.g., methylene region) and HMBC (e.g., amide carbonyl region) spectra, and data acquisition times for 2-D spectra can be substantially reduced by use of non-uniform sampling and NMR by ordered acquisition using ^1^H detection techniques.

NMR spectroscopy can also be used quantitatively to measure the concentrations of algal toxin solutions relative to reference solutions [116]. This approach has been applied to produce quantitative certified reference materials (CRMs) for MC-RR, MC-LR, and [Dha^7^]MC-LR [117] with CRMs for other MCs including MC-LA and [D-Leu^1^]MC-LY in preparation (P. McCarron, personal communication).

### 4.5. 3-Dimensional Structures of Microcystins

Two methods have been used for determining the 3-dimensional structures of MCs. X-ray crystallography has been used for nearly a century to reveal the 3-dimensional relationships of heavy atoms in crystallized organic molecules. Unfortunately, MCs have not so far yielded suitable crystals for such studies. However, in recent decades the 3-dimensional structures of MCs bound to protein phosphatases (PPs) have been determined [118,119,120,121,122,123,124,125,126] (see Section 6.2). In addition to providing key information about the mode of action and structure–activity relationships of the MCs, this also provides 3-dimensional structural information about the toxins themselves. The other approach that has been used is NMR spectroscopy, using through-space NOESY and ROESY correlations to estimate the relative intramolecular distances between hydrogen atoms in the MCs in solution [127,128,129,130]. Such measurements can be supplemented with measurements of ^1^H–^1^H coupling constants, which provide information about the dihedral angles between spin-coupled hydrogen atoms within the molecule. Such studies have led to several published 3-dimensional NMR-based solution structures, something that is not available from X-ray crystal structures because the structures are based on the solid-state structures that do not necessarily reflect the structure in solution.

Although not in and of itself proof of structure or stereochemistry, observation of high affinity of a purified MC for PPs or MC-specific antibodies can be taken as strong supporting evidence that the absolute stereochemistry of the MC is largely the same as that of other MCs. The reason is that recognition of MCs by receptor biomolecules is dependent on the 3-dimensional structure of the ligand (see Section 6.2), which is controlled by amino acids present in the MC and their stereochemistries. Biosynthetic reasoning also suggests that most MCs likely share the same stereochemistry, since they are assembled by closely related synthetases produced from genetically similar gene clusters, even in separate MC-producing genera (see Section 3).

## 5. Diversity of Characterized Microcystin Congeners

To date, the identification of at least 279 different MC congeners have been reported in the literature (Table 1) using various combinations of the techniques reviewed in Section 4. These congeners include MCs biosynthesized with structural variations at every amino acid position as well as the products of several chemical and biochemical transformations that can occur in the environment or the laboratory, all of which are reviewed in this section.

### 5.1. Congeners with Variable Amino Acid at Position 1

In the first ever report of the structural elucidation of a MC, MC-LA, produced by the South African *M. aeruginosa* strain WR70 (UV-0l0), the amino acid residue in position 1 (Figure 1) was found to be D-Ala [15,17,18,20]. This amino acid is incorporated by the NRPS module of *mcyA* [23] and was initially considered as relatively conserved in MCs [6,94]. However, subsequent characterization of new congeners has shown that the amino acid residue in position 1 is also variable, although not as diverse as those in positions 2 and 4. D-Ala is found in position 1 in 219 of the 279 reported MCs (Table 1). However, other amino acid residues reported in position 1 include D-Ser, D-Leu, Gly, D-Met, D-Phe, D-Val, D-Hil, and D-Met *S*-oxide. Among these congeners, the first one reported was [D-Ser^1^,ADMAdda^5^]MC-LR isolated from *Nostoc* sp. strain 152 by Sivonen et al. [86]. Subsequentlty, [D-Leu^1^]MC-LR was identified as the most abundant MC from a laboratory strain RST 9501 of a *Microcystis* sp. isolated from a hepatotoxic *Microcystis* bloom from brackish waters in the Patos Lagoon estuary, southern Brazil [167], and from a cyanobacterial bloom dominated by *M. aeruginosa* collected in Pakowki Lake, AB, Canada [168]. A decade later, Shishido et al. [75] identified [D-Met^1^]MC-LR and [D-Met^1^,D-Asp^3^]MC-LR from *M. aeruginosa* strains NPLJ-4 and RST 9501. Furthermore, the authors reported that the *M. aeruginosa* strain NPLJ-4 produced a larger diversity of MC variants, but most of them in trace amounts, such as [D-Val^1^]MC-LR and [D-Phe^1^]MC-LR. Recently, Puddick et al. [91] described eight new congeners with Gly at position-1 in two hydro-terrestrial cyanobacterial mat samples from the McMurdo Dry Valleys of Eastern Antarctica. Recently, Foss et al. [81] identified a new congener by LC-MS/MS with homoisoleucine (D-Hil) at position 1, from a *Microcystis* bloom at Poplar Island, MD, USA.

### 5.2. Congeners with Variable Amino Acid at Positions 2 and 4

The two L-amino acids in positions 2 and 4 are the most variable of the seven amino acids in MCs. The adenylation domains of *mcyB* and *mcyC* are responsible for the activation of the amino acids that appear in positions 2 and 4, respectively [23,54,55,56]. Therefore, the variable L-amino acids incorporated into MCs are dependent upon the substrate specificity of the adenylation domain responsible for loading the amino acid onto the MC synthetase module [32,179]. To date, 21 different amino acids have been identified at position 2 and 42 at position 4 (Table 1). Leu is present in position 2 in about one third of reported MCs (95 of the 279) and Arg is present in position 4 in more than half of the reported MCs (173 of the 279). For example, Puddick et al. [12], when conducting an analysis of the relevant literature on 49 isolated strains of cyanobacteria, reported that most of the strains assessed were only able to incorporate one or two amino acids into position 4, usually including Arg. Consequently, the congener MC-LR is the most commonly observed MC in most cyanobacterial blooms worldwide [6,7,8,9,180]. However, in many other cases, different congeners with differing residues at positions 2 and 4 can dominate a bloom and methods targeting MC-LR exclusively are known to significantly underestimate total MC content in many cases.

### 5.3. Congeners with Variable Amino Acid at Position 3

Position 3 is highly conserved in MCs, as only D-*erythro*-β-methyl isoaspartic acid (D-Masp^3^) and D-*erythro*-isoaspartic acid (D-Asp^3^) have been reported in MCs. D-Masp is most commonly reported (157 of 279 MCs), with D-Asp reported in all remaining MCs (Table 1). The amino acid at position-3 is incorporated by the activity of the NRPS module of *mcyB* [23]. Furthermore, Pearson et al. [181] and Sielaff et al. [182] reported that one of the moieties, D-*erythro*-β-methyl-aspartic acid, required the activity of two additional enzymes prior to incorporation in the peptide chain: the 2-hydroxy acid dehydrogenase *mcyI* and the aspartate racemase *mcyF*. Both dicarboxylic acids, D-Masp/Asp at position 3 and D-Glu in position 6, are inserted into the cyclic peptide structure in the *iso*-configuration [15,16].

### 5.4. Congeners with Variable Amino Acid at Position 5

The most characteristic twenty-carbon β-amino acid in position 5 of MCs, and which plays an important role in their biological activity, is Adda (reported in 220 of 279 MCs). Four PKS modules, *mcyG*, *J*, *D*, and *E* are responsible for the synthesis of the Adda moiety of MCs [24]. The natural structural diversity of the Adda moiety is largely limited to the group on the C-9 oxygen, that can be present as 9-*O*-desmethylAdda (DMAdda), the 9-hydroxy derivative of Adda, or as 9-acetoxy-9-*O*-desmethylAdda (ADMAdda). The *O*-methylation on the C-9 oxygen of MCs is carried out by the *mcyJ O*-methyltransferase [55]. Recently, Fewer et al. [183] reported that the production of *O*-acylated variants of MC in *Nostoc* is the result of specific enzymatic *O*-acetylation of the MC catalyzed by the *mcyL O*-acetyltransferase. In fact, several earlier studies reported that *O*-acylated MCs are produced almost exclusively by strains of the genus *Nostoc* isolated from a range of aquatic and terrestrial habitats in free-living and lichen-associated growth states [79,86,137,153]. However, *O*-acetylated MCs have also been reported in a single strain of *Planktothrix agardhii* PH123 [178].

The first DMAdda microcystin congener, [DMAdda^5^]MC-LR, was identified from a cyanobacterial bloom containing *M. aeruginosa* (dominant), *M. viridis* and *M. wesenbergii* collected from Homer Lake, IL, USA [76,80]. Furthermore, 15 different DMAdda MC congeners have been described (Table 1), that could originate from either incomplete methylation by the *mcyJ* gene during biosynthesis or from hydrolysis of the 9-methoxy or 9-acetoxy group in the [Adda^5^] and [ADMAdda^5^] MC variants, respectively. However, since most reports of DMAdda^5^-microcystins have occurred in strains or blooms where the corresponding ADMAdda-congeners were not present, and ethers are relatively resistant to hydrolysis, the origin of DMAdda^5^-microcystins is likely to be due primarily to incomplete methylation during biosynthesis. LC-MS studies recently indicated the existence of small amounts of incompletely characterized MC congeners containing oxidized Adda (hydroxyAdda and ketoAdda) and Adda demethylated (dmAdda) at two positions other than at the 9-*O*-methyl group [81]. Furthermore, [epoxyAdda^5^]MC-LR was tentatively identified in a field sample [81]. These oxidized congeners are likely to arise through metabolism or autoxidation, while the two dmAdda variants presumably arise through biosynthesis.

### 5.5. Congeners With Variable Amino Acid at Position 6

D-Glu is incorporated in position 6 by the activity of the NRPS module adjacent to the PKS module of *mcyE* [23] and is highly conserved in this position. The esterification of the α-carboxyl group of the D-Glu residue is rare, and only 11 such MC congeners have been characterized in the literature (Table 1). Namikoshi et al. [76,80] characterized the only non-methyl (-C_3_H_6_OH) monoester of the α-carboxyl on the D-Glu^6^ unit of MC-LR ([D-Glu(OC_2_H_3_(CH_3_)OH)^6^]MC-LR) from a toxic bloom containing *M. aeruginosa* (dominant), *M. viridis,* and *M. wesenbergii* collected from Homer Lake (Illinois) in 1988. However, the mono-methyl ester derivatives at the D-Glu^6^ unit [D-Glu(OMe)] were more frequent and 11 different congeners have been characterized (Table 1). These methyl esters appear to be artefacts produced by exposure to methanol (Section 5.7).

### 5.6. Congeners With Variable Amino Acid at Position 7

The amino acid residue in position 7 is also variable; however, it is not as diverse as those in positions 2 and 4. *N*-methyldehydroalanine (Mdha) is found in position 7 in 189 of the 279 reported MCs (Table 1). However, other amino acids have been reported in this position, including dehydroalanine (Dha) [33,83,108,109,138,161], dehydrobutyrine (Dhb) [75,77,79,91,142], *N*-methyldehydrobutyrine (Mdhb) [78], Ser [33,84,102,138,152,161], *N*-methylserine (Mser) [80,82,86,88,93,108,109], Thr [171], and conjugates formed by addition of Cys, γ-GluCys and glutathione (GSH) with the reactive double bond of Mdha^7^ [80,81]. Chemical and metabolic transformations of MCs are discussed in Section 5.7.

The *N*-terminal adenylation domain of *mcyA* catalyzes the *N*-methylation of MCs [21,23,55,56]. The *N*-methyltransferase (NMT) domain recognizes and activates Ser as an AMP-adenylate, causing the Ser residue to undergo *N*-methylation and dehydration to form Mdha [21,23,34,55,56]. Thus, the NMT domain catalyzes the transfer of the methyl group from *S*-adenosylmethionine (SAM) to the α-amino group of the thioesterified amino acid [184]. However, Fewer et al. [34] reported that the order and timing of the *N*-methylation, dehydration, and condensation reactions have not yet been determined. For example, deletion of the entire NMT domain of *mcyA* in *Anabaena* [34] and *Planktothrix* [185] or point mutations in this gene in *Microcystis* [37], were associated with the production of Dha^7^-microcystins by these genera. Therefore, simultaneous production of Mdha- and Dha-containing MCs in *Anabaena* could be the result of the inactivation of the NMT domain during the assembly of Dha-containing MCs [34]. However, Marahiel et al. [186] suggested that the concomitant biosynthesis of Dha- and Mdha-containing MC variants could also be the result of the limitation of SAM, the source of the *N*-methyl group. Such mixtures of MCs containing both Mdha and Dha are common in *Planktothrix* [114] and *Microcystis* [139], and partial inactivation of the NMT domain may be a widespread feature of MC biosynthesis.

MCs-containing Dhb instead of Dha in *Planktothrix* are thought to be the result of a homologous recombination event in the first module of *mcyA* that replaces the adenylation domain, leading to a change in substrate specificity (from Ser to Thr) and presumably the concomitant deletion of the NMT domain [185]. For the biosynthesis of nodularin, a microcystin-like pentapeptide containing Mdhb instead of Mdha, the nodularin synthetase gene cluster is proposed to have arisen by deletion of two NRPS modules and a change in substrate specificity from Ser to Thr in the C-terminal adenylation domain of *Nda*A [187]. Kurmayer et al. [185] reported a significant correlation between *mcyA*Ad1 genotypes with NMT and *mcyA*Ad1 genotypes without NMT and the occurrence of either Mdha or Dhb in position 7 of the MC structures in *Planktothrix* strains. Therefore, strains of cyanobacteria producing MC variants containing either Dha or Dhb, exclusively, are known from all major MC producing genera (*Microcystis*, *Nostoc*, *Anabaena*, and *Planktothrix*) [6,33,91,109,138,144]. Furthermore, Sano and Kaya [142] reported that the configuration of the Dhb^7^ unit of Dhb-microcystins isolated from the CCAP strains of *O. agardhii* was *E*, while the Dhb of nodularins from *Nodularia spumigena* has been determined as *Z* [188], suggesting that the Dhb unit in Dhb-microcystins is biosynthesized via a dehydration of Thr or *allo-*threonine.

The MC congener [D-Asp^3^,Thr^7^]MC-RR was identified from an environmental bloom of *P. rubescens* collected from Austrian alpine lakes [171]. Additionally, 11 MC analogues containing Ser and 13 containing Mser instead of Mdha in position 7 have been reported (Table 1). According to Tillett et al. [23], the module *mcyA* is responsible for the incorporation of Ser into the growing molecule. Ser is further *N*-methylated by the corresponding NMT domain and transformed by dehydration into Mdha prior to or following the condensation reaction. Analogous to Ser, Thr is transformed by dehydration into Dhb [189]. Therefore, the production of the unique congener with Thr residue and the congeners with Ser or Mser residues in position 7 could be the result of the partial inactivation of the NMT domain during the dehydration and *N*-methylation reactions of Thr- and Ser-containing MCs. In fact, Fewer et al. [34] reported that there is a 10–500-fold increase in the amount of Ser incorporated into the MCs in strains of *Anabaena* lacking a functional NMT domain, suggesting that there may be a functional coupling of the dehydration and *N*-methylation reactions.

### 5.7. Chemical and Biochemical Transformations of Microcystins

Several chemical and biochemical reactions have been reported to occur in the cell, the environment, or during sample extraction and storage in the laboratory that lead to the formation of new MC analogues beyond those that are biosynthetic products of cyanobacteria (Figure 2). Even though these are not of biosynthetic origin, their occurrence can still be important to measure for studies of MC occurrence and toxicity. It is recommended that authors describing the identification of new MC variants try to clearly differentiate between those that are metabolites of cyanobacteria, those that form through conjugation or metabolism after biosynthesis, and those that form artefactually during sample handling or storage.

Eight MCs containing sulfur oxidation in their Met moiety have been reported (Table 1) that are likely artefacts produced by autoxidation. This is consistent with the findings of Miles et al. [93] who reported that a fresh culture of *Dolichospermum flos-aquae* contained only traces of sulfoxides [D-Asp^3^]MC-M(O)R and MC-M(O)R, but that these increased markedly during storage or sample extraction and preparation. This suggests that MCs containing methionine sulfoxide are primarily post-extraction oxidation artefacts, rather than being produced by biosynthesis in cyanobacteria. These can then be further oxidized to the corresponding sulfones (e.g., MC-M(O_2_)R) through an additional step of autoxidation [93]. In addition, Puddick et al. [90] reported MC congeners containing the oxidized tryptophan moieties kynurenine (Kyn), oxindolylalanine (Oia), and *N*-formylkynurenine (Nfk) that are also considered as autoxidation artefacts.

Isomerization of the C-6 double bond in the Adda^5^ group is rarely reported and is thought to be due to ultraviolet (UV) irradiation rather than arising through biosynthesis [190,191,192]. In early studies, Harada et al. [115] and Namikoshi et al. [76] isolated geometrical isomers at C-6 of [(6*Z*)-Adda^5^]MC-LR and -RR, as minor components with along with the corresponding normal (6*E*)-Adda^5^ MCs (MC-LR and -RR) from *Microcystis* spp. The DMAdda derivative at the Adda unit could be also an artefact produced during extraction of cyanobacterial samples dominated by ADMAdda derivatives. In fact, Beattie et al. [150] observed that [ADMAdda^5^]MCs are unstable and hydrolyze at room temperature in aqueous solution and/or in methanol, and that [D-Asp^3^,ADMAdda^5^,Dhb^7^]MC-RR isolated from a *Nostoc* strain DUN901 was converted to [D-Asp^3^,DMAdda^5^,Dhb^7^]MC-RR within one week. Similarly, Ballot et al. [131] reported the half-life for hydrolysis of several [ADMadda^5^]MCs (to form the corresponding [DMadda^5^]MCs) to be ca 30 h at pH 9.7 and 30 °C.

In addition, MCs containing methyl esters in their Glu^6^ moiety are almost certainly artefacts produced by esterification in methanolic solutions, possibly due to traces of residual acid from extraction or isolation steps. For example, [D-Glu(OMe)^6^]MC-LR and [D-Glu(OMe)^6^]MC-FR characterized recently by Bouhaddada et al. [40] were identified in the presence of their much more abundant non-esterified congeners MC-LR and MC-FR, respectively. In addition, the same results were reported for the eight other described congeners with methyl esters on the α-carboxyl group of the D-Glu residue in position 6 [76,80,85,95,152,169]. Harada et al. [112] demonstrated formation of Glu^6^ methyl esters of MCs in acidic methanolic solutions, and Foss et al. [81] observed substantial esterification of [Leu^1^]MC-LR that was purified with an acidic eluent then evaporated to dryness before storage in MeOH at −20 °C. In fact, del Campo and Ouahid [38] identified the presence of [D-Asp^3^,D-Glu(OMe)^6^]MC-LAba from a cultured *Microcystis* strain UTEX-2666, but not its non-esterified residue [D-Asp^3^]MC-LAba, suggesting that near-complete esterification is possible under some conditions. Therefore, the technique described by Fastner et al. [193] consisting of analyzing of a single colony of *Microcystis* directly, without methanol extraction may circumvent this probem. As a precaution, use of both acids and methanol during sample preparation or storage should be avoided where possible. Esters of MCs containing L-Glu at position-2 and -4 have also been reported [138] and given the facile esterification of D-Glu^6^ it seems probable that these two are esterification artefacts as well.

MCs with either Dha or Mdha at position 7 possess an α,β-unsaturated carbonyl group that is reactive toward thiols (e.g., cysteine), and to a lesser extent hydroxyl groups, through the Michael addition reaction [100,101]. Consequently, the unusual MC congener [Mlan^7^]MC-LR that was characterized from a bloom of *Microcystis* spp. collected from Homer Lake (IL, USA) can be considered the result of a chemical reaction between the major constituent (MC-LR) biosynthesized in the cyanobacterium and cysteine [80]. On the other hand, the biosynthesis of lanthionine has been proposed to involve addition of the thiol group of a Cys residue to the double bond of Dha, generated by dehydration from Ser, forming a sulfide bridge [80]. It has also been proposed that glutathione-*S*-transferase catalyzes the addition of glutathione (GSH) to MCs, but also that the reaction proceeds at a similar rate with or without the enzyme under physiological conditions [101,194], suggesting that the reaction may be primarily chemical rather than purely biochemical. Recently, 3 new MCs containing cysteine (Cys), glutathione (GSH), and γ-glutamylcysteine (γ-GluCys) conjugated to the Mdha^7^ olefinic bond of [D-Leu^1^]MC-LR were identified in extracts of freeze-dried *Microcystis* bloom material from Poplar Island (MD, USA) [81]. Small amounts of the corresponding sulfoxides of the conjugates (Cys(O), cysteine sulfoxide; GS(O), glutathione sulfoxide; and γ-GluCys(O), γ-glutamylcysteine sulfoxide), probably produced by autoxidation, were also detected. Conjugated MCs present in the bloom material were characterized for the first time, but they could nonetheless be metabolites produced during detoxification of MC congeners in aquatic organisms. The principal pathway of the metabolism and the excretion of MCs in mammals and aquatic organisms has been found to be as MC–GSH and MC–Cys conjugates [8,195,196,197]. However, cyanobacteria are known to produce both Cys and GSH [198], and low levels of GSH conjugates of MCs have been detected in several cyanobacterial cultures using untargeted LC-HRMS methods (D. G. Beach and C. O. Miles, unpublished observations; [99]). GSH conjugates can be further metabolized to γ-GluCys and Cys conjugates [199], so it is possible that the γ-GluCys and Cys conjugates observed from cyanobacteria are produced via this route or by direct conjugation of MCs to these smaller peptides. Nucleophilic addition of water (to produce D,L-Ser^7^ and -Mser^7^ congeners) and methanol (to produce the corresponding *O*-methoxyserine congeners), and intramolecular addition of the 2-arginyl group, have all also been observed at the reactive double bond of Dha^7^ and Mdha^7^ of MCs under basic conditions during studies of their derivatization with thiols [101], as shown in Figure 2.

Eight linear MCs (Table 1) have been reported to date that could well be the products of bacterial degradation of MCs, since many studies have shown that the first step of biodegradation of MCs by multiple different species of bacteria is through the cleavage and opening of the macrocycle (reviewed by Li et al. [200]).

MC congeners containing oxidized Adda (an epoxide, ketone, and hydroxylated derivative) have also been reported recently in small amounts in a bloom, and partially characterized by LC-HRMS/MS and chemical derivatizations [81]. At this stage it is unclear whether these are the result of enzyme-catalyzed oxidation or are products of autoxidation.

Researchers should consider the detection in a sample of a potentially artefactual MC congener, such as an oxidized, esterified, or conjugated variant, as possible evidence for the presence in the original sample of its unmodified precursor MC.

## 6. Toxicity of Microcystin Congeners

### 6.1. Toxicity In Vivo

It should be borne in mind when comparing LD_50_ values (the dose of toxin that kills 50% of exposed animals) that toxicological measurements are inherently subject to uncertainty caused by biological variability, and the possibility that in some cases isolated MCs may not have been adequately characterized with respect to purity and amount of toxin prior to toxicity assessment. For this reason, only large differences in toxicity should be considered significant when comparing toxicities from different experiments and different MC congeners.

The toxic effects of MCs on animals were first studied with unpurified cyanobacterial extracts. Several studies reported enlargement and congestion of the liver, with necrosis of the hepatic cells in laboratory and domestic animals after oral administration or parenteral injection of toxic algal extracts in which *M. aeruginosa* was the predominant species [44,201,202]. A decade later, Elleman et al. [166] confirmed these observations and reported that the lethal dose (LD_100_) in mice by intraperitoneal (i.p.) injection of an *M. aeruginosa* (then *Anacystis cyanea*) bloom extract was 15–30 mg of lyophilized algal bloom per kilogram body weight (b.w.). The LD_50_ of the purified toxin from this extract was estimated to be 56 µg kg^−1^ in mice, and the approximate LD_100_ was 70 µg kg^−1^. Acute toxicity by parenteral administration of the purified toxin to mice produced extensive liver lobular hemorrhage and death within 1–3 h. However, repeated administration of sublethal doses daily over some weeks produced progressive hepatocyte degeneration and necrosis and the development of fine hepatic fibrosis.

Since then and following the identification of the toxin responsible, toxicological research has focused on purified MCs. The most extensive toxicological information is available for MC-LR. The acute toxicity of MC-LR has been tested in several studies on mice, with i.p. LD_50_ values ranging from 25 to 150 µg kg^−1^ b.w. [203,204]. An LD_50_ value of 50 µg kg^−1^ b.w. is generally accepted (Table 2) (for a review, see [205]). The inhalation toxicity (intratracheal and intranasal) in mice is similar, with an LD_50_ of 36 to 122 µg kg^−1^ b.w. [206,207,208,209]. However, MC-LR is 100 times less toxic by oral ingestion than it is by intra-peritoneal injection, with an LD_50_ of 5000 µg kg^−1^ b.w. [203,210]. Although some studies indicate that MCs are stable to proteolytic enzymes in vitro [211], Freitas et al. [212] suggested that a decrease in the bioaccessibility and thus bioavailability of free MC-LR after proteolytic digestion of contaminated shellfish by the pancreatic enzymes, trypsin and chymotrypsin, may be an important factor that can decrease its toxicity by oral route. Although the toxicity of all characterized MC congeners has not been determined, there are some variations within the MC family with i.p. LD_50_ values ranging from 50 to >1200 µg kg^−1^ b.w. in mice (see Table 2). Firstly, the cyclic structure of MCs is very important for the toxicity and opening of the macrocycle at any position makes the molecule much less toxic. For example, the two congeners [seco-1/2]MC-LR and [seco-4/5]MC-LR, corresponding to the opening of the cyclic structure of the congener MC-LR between the amino acid residues at positions 1/2 and 4/5, respectively, showed no toxicity to mice at 22,500 and 1100 µg kg^−1^ b.w., respectively [80]. Secondly, two of the most conserved residues, at position 5 (Adda, present in 220 of 279 identified MC congeners) and position 6 (D-Glu, present in 267 of 279 identified MC congeners) are also very important for the toxicological potential. Adda group stereochemistry and its double bond configuration (6*E*) are crucial for the toxicity since [(6*Z*)-Adda^5^]MC-LR showed an LD_50_ in mice >1200 µg kg^−^^1^ b.w. [115,147]. However, the toxicity appears to be only slightly affected by 9-*O*-demethylation (DMAdda) or 9-*O*-acetylation (ADMAdda) of the Adda group. [DMAdda^5^]MC-LR and [ADMAdda^5^]MC-LR were found to have LD_50_ values of 90–100 and 60 µg kg^−1^ b.w., respectively, by i.p. injection in mice. The free carboxylic acid on the D-Glu^6^ residue seems to be essential for toxicity because the esters ([D-Glu(OC_3_H_7_O)^6^]MC-LR [76] and [D-Glu(OMe)^6^]MC-LR) [85,140,189] of the MC-LR showed no toxicity in the i.p. mouse bioassay at 1000 µg kg^−1^ b.w. These results are consistent with the results of studies with synthetic fragments containing elements of the MC structure, where the presence of Adda and a suitably placed carboxylic acid group mimicking the carboxyl group of Glu^6^ was enough to elicit PP inhibition, albeit at potencies far below that of MC-LR [213,214]. However, substitution of amino acids in positions 1, 3, and 7 only had a small impact on the toxic potential of MCs in vivo. For example, MC toxicity seems to be barely affected by substitution of the moderately conserved D-Ala in position 1 by other D-amino acids. Matthiensen et al. [167] and Park et al. [168] reported that the LD_50_ of [D-Leu^1^]MC-LR by i.p injection in mice was 100 µg kg^-1^ b.w. Likewise, MC toxicity was only slightly affected by demethylation of the amino acid at position 3 (D-Asp^3^ instead of D-Masp^3^) or by demethylation or amino acid substitution at position 7. For example, demethylated forms of MCs containing D-Asp^3^ or Dha^7^ residues are two to five times less or more toxic, respectively, compared to their methylated forms [83,139,215,216]. Similarly, Sano and Kaya [142] reported that the toxicity is slightly affected by the addition of a methyl group to the double bond of the Dha^7^ unit to form the Dhb^7^, whereas, conjugation of the Mdha^7^ residue with Cys residue (to give [L-Mlan^7^] instead of Mdha^7)^) decreased the toxicity 20-fold compared to MC-LR by i.p. injection in mice [80]. Nevertheless, the acute toxicity is significantly affected by substitution at the two most variable amino acid residues in positions 2 and 4. Most of the structural MC congeners containing a hydrophobic L-amino acid in position 2 or 4, such as MC-LA, MC-YR, MC-YA, and MC-YM(O), have similar LD_50_ values to MC-LR (Table 2), whereas the LD_50_ of the hydrophilic MC-RR is about ten times higher (Table 2).

Most of the differences in in vivo toxicity between MC congeners can be attributed to a combination of differences in their uptake into the liver via a carrier-mediated transport system (organic anion transporting polypeptides (OATPs)) and/or their inhibitory potency toward hepatic PPs that are widely recognized as the principal targets of MCs [95,204,217,218,219,220,221,222,223]. MC toxicity requires active transport by OATP1B1 and OATP1B3, members of the OATP family of transporters, for uptake into cells and the high expression of these transporters in the liver accounts for their selective hepatotoxicity [95,224]. While some studies suggest that differences in in vivo and in vitro toxicity are related to PP inhibition rather than transport [216,225], other reports show that in vivo toxicity depends mainly on OATP transport kinetics rather than on differences in PP inhibition [215,224]. Therefore, future study is required to determine the relative importance/contribution of OATP versus PP inhibition to overall toxicity of MCs.

### 6.2. In Vitro Toxicity: Inhibition of Serine/Threonine PPs

The nature of the interaction of MCs with serine/threonine PPs has been intensively studied [118,217,221,228,229,230,231]. The concentrations required to inhibit 50% of the enzyme activity (IC_50_) of serine/threonine phosphatases (PP1–6) by MC congeners are shown in Table 3. MC-LR, the most toxic congener, has similar potency in the inhibition of PP1, PP2, and PP4–6, presenting an IC_50_ of around 0.1–1 nM [232,233,234,235]. However, it is 1000-fold less potent towards PP3 (IC_50_ ca. 1 µM) and seems to have no effect on the activity of PP7 [235,236].

The inhibition of the PP activity by MCs results primarily from non-covalent interactions largely mediated by the hydrophobic Adda side-chain in position 5, and the Glu^6^ moiety [229,255,256]. The Adda side-chain is closely accommodated in the hydrophobic groove of the catalytic subunit of the enzyme (Figure 3) and accounts for a significant portion of the toxin’s binding potential. Xing et al. [119] reported that in the catalytic subunit of PP2A, four amino acids Gln122, Ile123, His191, and Trp200 form a hydrophobic cage that accommodates the long hydrophobic Adda side-chain in MC-LR (Figure 3). The second important interaction consist of those of the carboxylic acid of the Glu^6^ residue and its adjacent carbonyl group (Adda-amide), which make hydrogen bonds to metal-bound water molecules in the active site of the catalytic subunits of PPs (Figure 3). These obsevations from X-ray crystallography agree well with in vitro and in vivo toxicity data that found Glu and Adda residues to be essential for the toxicity of MCs, and that modifications to the configuration of the Adda side-chain and esterification of the free carboxylic acid of Glu^6^ both lead to loss of toxic potential [76,255,256].

A third interaction also occurs between MCs and the catalytic sub-units of PPs via covalent linkage of the electrophilic α,β-unsaturated carbonyl of the Mdha residue in position 7 to the thiol of the cysteine residue in the catalytic subunits of six serine⁄threonine-specific phosphatases (PP1, PP2, PP3, PP4, PP5, and PP6) by Michael addition (reviewed in [256]). Interactions in the binding pocket between PP1 [118,228] and PP2A [119] and MC-LR are strengthened by a covalent linkage between the Sγ atom of Cys-273 and Cys-269, respectively and the terminal carbon atom (C-3) of the Mdha side chain. However, formation of the covalent bond occurs slowly, and is not essential for the inactivation of the PPs by MCs [100,228,229,251]. For example, Miles et al. [108,109] have shown that Dhb^7^-containing MCs react with thiols about 200 times more slowly than Mdha^7^ and Dha^7^-containing MCs [108]. Nodularin-R, which is structurally related to MC-LR but is a pentapeptide containing *N*-methyldehydrobutyrine (Mdhb) at position-5, did not react at a detectable rate with mercaptoethanol [108]. However, nodularin-R exhibits an IC_50_ (1.8 nM), like that of MC-LR (2.2 nM) for human red blood cells PP2A [250] and nodularins do not form a covalent linkage to thiol group of Cys-273 in PP1 [120,258]. Similarly, MC congeners containing Dha^7^ (e.g., [D-Asp^3^,Dha^7^]MC-LR) and its analogue containing Dhb^7^ ([D-Asp^3^,Dhb^7^]MC-LR) have similar IC_50_ values for PP2A of 0.254 ± 0.004 and 0.201 ± 0.003 nM, respectively [239].

Two other interactions that also play an important role in the inactivation of PPs have been recently described [256]. The first consists of a hydrogen-bonding interaction between the carboxyl group of the Masp^3^ residue to the conserved Arg and Tyr residues in PP’s sub-catalytic units (Arg96 and Tyr134 of PP1; and Arg89 and Tyr127 of PP2). The second consists of a hydrophobic interaction between the aromatic ring of a conserved tyrosine (Tyr272 of PP1; and Tyr265 of PP2) in the PP’s β12–β13 loop to the L-Leu^2^ residue of MC-LZ congeners [256]. The presence of L-leucine in position 2 can, therefore, explain the higher PP-inhibition potential of MC-LR and its analogues with hydrophobic amino acids in the same position (Table 3).

## 7. Conclusions and Future Perspectives

Many MCs have been isolated and fully characterized over the last three decades using a combination of analytical techniques and advances in LC-MS/MS instrumentation have recently provided a rapid influx of new MC congeners by tentative identification of trace variants in complex samples. As described in this review, multiple combinations of the variable L-amino acids at positions 2 and 4 and modifications to the other amino acids lead to the structural diversity of 279 MC variants characterized to date. These are formed because of genetic diversity in the *mcy* gene cluster as well as subsequent chemical and biochemical transformations that can occur after biosynthesis. Although the discovery of *myc* gene clusters has led to new avenues for investigating MC regulation at the molecular level, gaps remain in the understanding of how environmental factors can modulate the relative abundance of MC congeners in a bloom. Consequently, because the toxicity of a cyanobacterial bloom depends not only on the total MC accumulation, but also on the structure and the abundance of each congener, further research focusing on the effects of environmental factors on modulating the MC profile during bloom episodes is therefore recommended. This is particularly important in the context of new public health guidelines emerging around the world based on ‘total microcystins’, which remains a poorly defined concept in the absence of comprehensive data on the strucutural diversity and toxic potential of the broader class.

Variations in in vivo toxicity between MC congeners can be attributed to the relative importance/contribution of OATP transport versus PP inhibition to overall toxicity of MCs. However, the large range of values reported in the literature for the LD_50_ by the same route of exposure within each congener can in part be explained by a combination of biological variability as well as uncertainty in toxin purity and quantitation. It is, therefore, very important for researchers to carefully assess and report on the purity and amount of toxins used for toxicity assessments to ensure comparability of results between different studies and different MC variants.

## Figures and Tables

**Figure 1 toxins-11-00714-f001:**
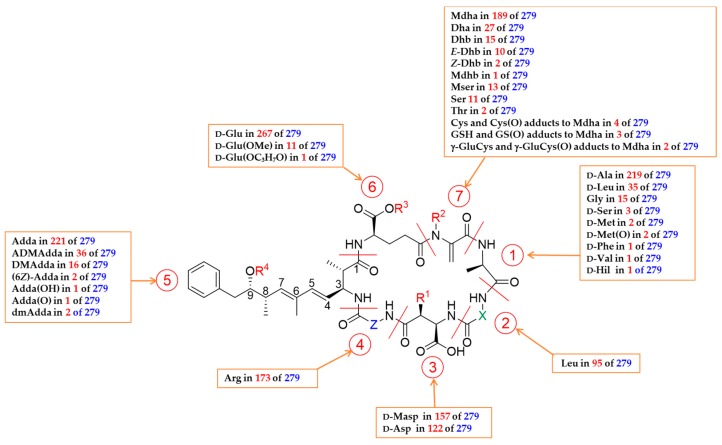
General structure of microcystins (MCs) and an overview of their observed structural diversity. R^1^ = H or CH_3_; R^2^ = H or CH_3_; R^3^ = H, CH_3_ or C_3_H_6_OH; R^4^ = H, CH_3_ or COCH_3_; X and Z = variable L-amino acids.

**Figure 2 toxins-11-00714-f002:**
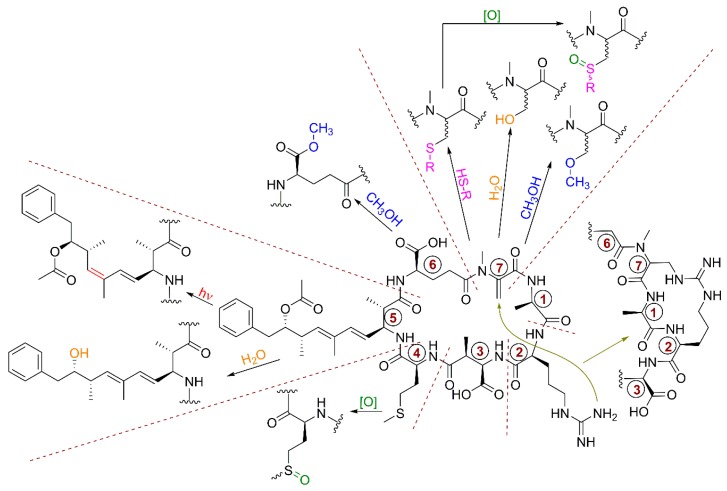
Examples of chemical reactions of MCs demonstrated using the hypothetical congener [ADMdda^5^]MC-RM. The numbers in circles indicate the amino acid residue-number.

**Figure 3 toxins-11-00714-f003:**
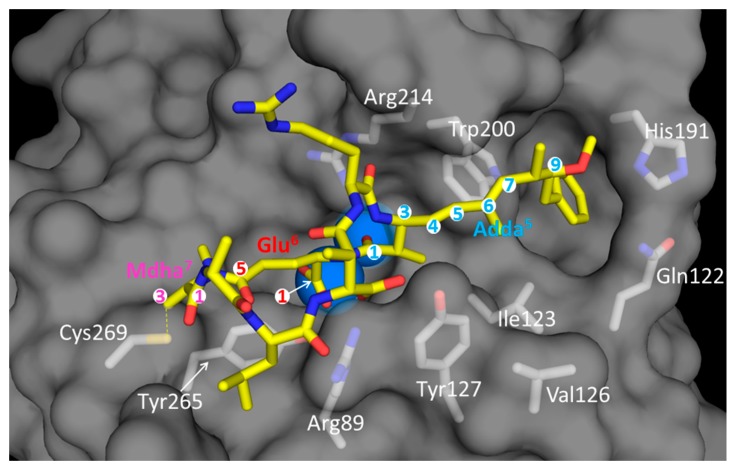
X-ray crystallographic structure showing the interaction of microcystin-LR (MC-LR) with the PP2A catalytic subunit and its adjacent amino acid side chains (based on data provided in [257]). Important interactions are described in the text (Section 6.2). The Mdha^7^, D-Glu^6^ and Adda^5^ moities, and selected atoms in these amino acids, are labelled in magenta, red, and pale blue, respectively. The large blue spheres are the catalytic metal ions, and the dashed yellow line shows the location of the covalent bond formed over time between the thiol of Cys269 and the Mdha^7^ moiety of MC-LR.

**Table 1 toxins-11-00714-t001:** Microcystin congeners reported in the literature as identified from cyanobacterial cultures and field samples.

Entry	Microcystin	Molecular Formula	Exact Mass ^a^	Characterization	Reference
1	[D-Asp³,DMAdda^5^]MC-LA	C_44_H_63_N_7_O_12_	881.4535	LC-MS/MS	[33]
2	[D-Asp³]MC-VA	C_44_H_63_N_7_O_12_	881.4535	LC-MS/MS	[33]
3	[D-Asp^3^]MC-LA	C_45_H_65_N_7_O_12_	895.4691	LC-HRMS/MS	[94]
4	[Dha^7^]MC-LA	C_45_H_65_N_7_O_12_	895.4691	LC-MS/MS	[33]
5	[DMAdda^5^]MC-LA	C_45_H_65_N_7_O_12_	895.4691	LC–MS/MS, thiol	[131]
6	MC-VA	C_45_H_65_N_7_O_12_	895.4691	LC-MS/MS	[33]
7	MC-LA	C_46_H_67_N_7_O_12_	909.4848	MS, NMR, AA, LC-HRMS/MS	[15,94]
8	MC-LAbu	C_47_H_69_N_7_O_12_	923.5004	HRFABMS, AA, NMR	[132]
9	[D-Asp^3^,D-Glu(OMe)^6^]MC-LAbu ^b^	C_47_H_69_N_7_O_12_	923.5004	LC-HRMS/MS	[38]
10	[D-Asp³]MC-LV	C_47_H_69_N_7_O_12_	923.5004	LC-MS/MS	[33]
11	[D-Asp³]MC-FA	C_48_H_63_N_7_O_12_	929.4535	LC-MS/MS, thiol	[12]
12	[D-Asp³,Dha^7^]MC-YA	C_47_H_61_N_7_O_13_	931.4327	MALDI-TOF MS	[89]
13	[D-Asp^3^]MC-LL	C_48_H_71_N_7_O_12_	937.5161	LC-HRMS/MS	[94]
14	MC-LV	C_48_H_71_N_7_O_12_	937.5161	AA, LC-MS/MS	[33,133]
15	[D-Asp³]MC-RA	C_45_H_66_N_10_O_12_	938.4862	LC-MS/MS, thiol	[12,33]
16	MC-FA	C_49_H_65_N_7_O_12_	943.4691	NMR, LC-MS/MS, AA, thiol	[134]
17	MC-LL	C_49_H_73_N_7_O_12_	951.5317	LC-HRMS/MS, AA	[94,133]
18	MC-AR	C_46_H_68_N_10_O_12_	952.5018	AA, HRMS, NMR	[76]
19	MC-RA	C_46_H_68_N_10_O_12_	952.5018	LC-MS/MS, NMR, thiol	[77,135]
20	[D-Asp³]MC-RAbu	C_46_H_68_N_10_O_12_	952.5018	LC-MS/MS, thiol	[12]
21	MC-FAbu	C_50_H_67_N_7_O_12_	957.4848	LC-MS/MS, thiol	[12]
22	MC-YA	C_49_H_65_N_7_O_13_	959.4640	AA, NMR, MS, LC-MS/MS, thiol	[16,77]
23	MC-AHar	C_47_H_70_N_10_O_12_	966.5175	LC-MS/MS	[136]
24	[Gly^1^,D-Asp^3^]MC-LR	C_47_H_70_N_10_O_12_	966.5175	LC-MS/MS	[92]
25	[D-Asp^3^,Dha^7^]MC-LR	C_47_H_70_N_10_O_12_	966.5175	AA, FABMS/MS	[84]
26	[D-Asp³,DMAdda^5^]MC-LR	C_47_H_70_N_10_O_12_	966.5175	LC-MS/MS	[137]
27	[D-Asp³,DMAdda^5^,Dhb^7^]MC-LR	C_47_H_70_N_10_O_12_	966.5175	LC-MS/MS, thiol	[109]
28	[Gly¹,D-Asp^3^,Dhb^7^]MC-LR	C_47_H_70_N_10_O_12_	966.5175	LC-MS/MS, HRMS, AA, thiol	[91]
29	MC-RAbu	C_47_H_70_N_10_O_12_	966.5175	LC-MS/MS, thiol	[77]
30	[D-Asp^3^]MC-HarAbu	C_47_H_70_N_10_O_12_	966.5175	LC-MS/MS	[40]
31	[D-Asp³]MC-WA	C_50_H_64_N_8_O_12_	968.4644	LC-MS/MS, thiol	[12]
32	[D-Asp^3^,Dha^7^]MC-EE(OMe) ^b^	C_46_H_63_N_7_O_16_	969.4331	HRMS, MS/MS	[138]
33	[D-Asp³]MC-EE	C_46_H_63_N_7_O_16_	969.4331	LC-HRMS/MS, thiol, esterification, ^15^N-label	[99]
34	MC-LM	C_48_H_71_N_7_O_12_S	969.4881	AA, MS	[133]
35	[D-Asp^3^]MC-LF	C_51_H_69_N_7_O_12_	971.5004	LC-MS, MS/MS, LC-MS/MS, thiol	[108,139]
36	MC-VF	C_51_H_69_N_7_O_12_	971.5004	LC-MS/MS, ^15^N-enrichment	[140]
37	[D-Asp^3^,Dha^7^]MC-LY	C_50_H_67_N_7_O_13_	973.4797	LC-MS/MS, thiol	[108]
38	MC-YAbu	C_50_H_67_N_7_O_13_	973.4797	LC-MS/MS, thiol	[77]
39	[D-Asp^3^]MC-LR	C_48_H_72_N_10_O_12_	980.5331	MS/MS, HRMS, AA	[141]
40	[D-Asp^3^,(*E*)-Dhb^7^]MC-LR	C_48_H_72_N_10_O_12_	980.5331	NMR, AA, HRMS	[142]
41	[D-Asp^3^,(*Z*)-Dhb^7^]MC-LR	C_48_H_72_N_10_O_12_	980.5331	NMR, AA, HRMS	[142]
42	[Dha^7^]MC-LR	C_48_H_72_N_10_O_12_	980.5331	AA, FABMS/MS	[84]
43	[DMAdda^5^]MC-LR	C_48_H_72_N_10_O_12_	980.5331	AA, HRMS, NMR	[76]
44	[Gly^1^,D-Asp^3^,Dhb^7^]MC-LHar	C_48_H_72_N_10_O_12_	980.5331	LC-MS/MS, HRMS, AA, thiol	[91]
45	MC-RApa	C_48_H_72_N_10_O_12_	980.5331	LC-MS/MS, thiol	[77]
46	MC-VR	C_48_H_72_N_10_O_12_	980.5331	LC-MS/MS	[143]
47	MC-WA	C_51_H_66_N_8_O_12_	982.4800	NMR, LC-MS/MS, AA, thiol	[134]
48	[D-Ser^1^,D-Asp³,Dha^7^]MC-LR	C_47_H_70_N_10_O_13_	982.5124	LC-MS/MS	[33]
49	[Dha^7^]MC-EE(OMe) ^b^	C_47_H_65_N_7_O_16_	983.4488	HRMS, MS/MS	[138]
50	[D-Asp^3^,Dha^7^]MCE(OMe)E(OMe) ^b^	C_47_H_65_N_7_O_16_	983.4488	HRMS, MS/MS	[138]
51	MC-FL	C_52_H_71_N_7_O_12_	985.5161	LC-MS/MS, thiol	[12]
52	MC-LF	C_52_H_71_N_7_O_12_	985.5161	AA, MS	[133]
53	MC-KynA ^b^	C_50_H_66_N_8_O_13_	986.4749	LC-MS/MS, HRMS, thiol, semisynthesis	[90]
54	[D-Asp³]MC-LY	C_51_H_69_N_7_O_13_	987.4953	LC-MS/MS, thiol	[108]
55	[D-Asp^3^,(*E*)-Dhb^7^]MC-LY	C_51_H_69_N_7_O_13_	987.4953	NMR, LC-HRMS/MS, thiol	[109]
56	[Gly^1^,D-Asp^3^,ADMAdda^5^]MC-LR	C_48_H_70_N_10_O_13_	994.5124	LC-MS/MS	[92]
57	[Gly^1^,D-Asp³,ADMAdda^5^,Dhb^7^]MC-LR	C_48_H_70_N_10_O_13_	994.5124	LC-MS/MS, HRMS, AA, thiol	[91]
58	[D-Asp³,ADMAdda^5^]MC-VR	C_48_H_70_N_10_O_13_	994.5124	LC-MS/MS	[33]
59	[D-Asp³,Dhb^7^]MC-AhaR	C_49_H_74_N_10_O_12_	994.5488	LC-MS/MS, thiol	[109]
60	[D-Asp³]MC-Hil/HleR	C_49_H_74_N_10_O_12_	994.5488	LC-MS/MS	[33]
61	[D-Asp³,(*E*)-Dhb^7^]MC-HilR	C_49_H_74_N_10_O_12_	994.5488	NMR, HRMS, AA	[144]
62	[Dha^7^]MC-HilR	C_49_H_74_N_10_O_12_	994.5488	HRMS, NMR, AA	[145]
63	[DMAdda^5^]MC-HilR	C_49_H_74_N_10_O_12_	994.5488	LC-MS/MS	[33]
64	[DMAdda^5^]MC-LHar	C_49_H_74_N_10_O_12_	994.5488	LC-MS/MS	[33]
65	[D-Asp^3^,D-Glu(OMe)^6^]MC-LR ^b^	C_49_H_74_N_10_O_12_	994.5488	HRMS, MS/MS, AA	[85]
66	MC-LR	C_49_H_74_N_10_O_12_	994.5488	AA, NMR, HRMS, LC-MS/MS	[16,141,146]
67	[D-Asp^3^]MC-ER	C_47_H_68_N_10_O_14_	996.4916	LC-HRMS/MS, thiol, esterification, ^15^N-label	[99]
68	[(6*Z*)-Adda^5^]MC-LR ^b^	C_49_H_74_N_10_O_12_	994.5488	NMR, AA, MS	[115,147]
69	MC-RL	C_49_H_74_N_10_O_12_	994.5488	LC-MS/MS, thiol	[77]
70	MC-WAbu	C_52_H_68_N_8_O_12_	996.4957	LC-MS/MS, thiol	[90]
71	[Dha^7^]MC-E(OMe)E(OMe) ^b^	C_48_H_67_N_7_O_16_	997.4644	HRMS, MS/MS	[138]
72	MC-OiaA ^b^	C_51_H_66_N_8_O_13_	998.4749	LC-MS/MS, HRMS, thiol, semisynthesis	[90]
73	[D-Asp³]MC-MR	C_47_H_70_N_10_O_12_S	998.4895	LC-MS/MS, thiol, *S*-oxidation	[93]
74	[seco-4/5][D-Asp³]MC-LR ^b^	C_48_H_74_N_10_O_13_	998.5437	LC-MS/MS, thiol	[93]
75	[D-Asp³,Mser^7^]MC-LR	C_48_H_74_N_10_O_13_	998.5437	LC-MS/MS, MS/MS, thiol	[88,93]
76	[Ser^7^]MC-LR	C_48_H_74_N_10_O_13_	998.5437	AA, HRMS, MS/MS	[84]
77	MC-LHph	C_53_H_73_N_7_O_12_	999.5317	LC-MS/MS	[75]
78	MC-KynAbu ^b^	C_51_H_68_N_8_O_13_	1000.4906	LC-MS/MS, thiol	[90]
79	[D-Asp³,Dha^7^]MC-FR	C_50_H_68_N_10_O_12_	1000.5018	LC-MS/MS	[33]
80	[Ser^7^]MC-EE(OMe) ^b^	C_47_H_67_N_7_O_17_	1001.4593	HRMS, MS/MS	[138]
81	[D-Asp^3^,Ser^7^]MC-E(OMe)E(OMe) ^b^	C_47_H_67_N_7_O_17_	1001.4593	HRMS, MS/MS	[138]
82	[D-Asp³]MC-HilY	C_52_H_71_N_7_O_13_	1001.5110	LC-MS/MS, thiol	[108]
83	MC-LY	C_52_H_71_N_7_O_13_	1001.5110	LC-MS/MS, NMR	[38,148]
84	MC-YL	C_52_H_71_N_7_O_13_	1001.5110	LC-MS/MS	[38]
85	[D-Asp³,Mser^7^]MC-LY	C_51_H_71_N_7_O_14_	1005.5059	LC-MS/MS, thiol	[108]
86	[D-Asp³,ADMAdda^5^,Dha^7^]MC-HilR	C_49_H_72_N_10_O_13_	1008.5280	LC-MS/MS	[33]
87	[Gly^1^,D-Asp^3^,ADMAdda^5^,Dhb^7^]MC-LHar	C_49_H_72_N_10_O_13_	1008.5280	LC-MS/MS, HRMS, AA, thiol	[91]
88	[Gly^1^,D-Asp^3^,ADMAdda^5^]MC-LHar	C_49_H_72_N_10_O_13_	1008.5280	LC-MS/MS	[92]
89	[D-Asp^3^,ADMAdda^5^]MC-LR	C_49_H_72_N_10_O_13_	1008.5280	HRMS, NMR, AA, MS/MS, LC-MS/MS	[78,79,149]
90	[ADMAdda^5^,Dha^7^]MC-LR	C_49_H_72_N_10_O_13_	1008.5280	LC-MS/MS	[33]
91	[D-Asp³,ADMAdda^5^,Dhb^7^]MC-LR	C_49_H_72_N_10_O_13_	1008.5280	NMR, HRMS, AA	[150]
92	MC-HilR	C_50_H_76_N_10_O_12_	1008.5644	MS/MS, HRMS, NMR, AA	[80]
93	MC-LHar	C_50_H_76_N_10_O_12_	1008.5644	AA, MS/MS, HRMS, NMR	[151]
94	[D-Glu(OMe)^6^]MC-LR ^b^	C_50_H_76_N_10_O_12_	1008.5644	HRMS, MS/MS, AA, HRMS/MS	[85,152]
95	[Mdhb^7^]MC-LR	C_50_H_76_N_10_O_12_	1008.5644	AA, MS	[79]
96	[D-Leu^1^,D-Asp³,DMAdda^5^]MC-LR	C_50_H_76_N_10_O_12_	1008.5644	LC-MS/MS	[153]
97	[D-Asp^3^,Dha^7^]MC-RR	C_47_H_71_N_13_O_12_	1009.5345	AA, NMR, HRMS	[76]
98	[D-Asp³,DMAdda^5^]MC-RR	C_47_H_71_N_13_O_12_	1009.5345	MS, MS/MS	[55]
99	[Gly^1^,D-Asp^3^]MC-RR	C_47_H_71_N_13_O_12_	1009.5345	LC-MS/MS	[92]
100	[Gly^1^,D-Asp^3^,Dhb^7^]MC-RR	C_47_H_71_N_13_O_12_	1009.5345	LC-MS/MS, HRMS, AA, thiol	[91]
101	[D-Asp^3^]MC-LW	C_53_H_70_N_8_O_12_	1010.5113	LC-MS, MS/MS	[139]
102	[D-Asp^3^,(*E*)-Dhb^7^]MC-LW	C_53_H_70_N_8_O_12_	1010.5113	NMR, LC-HRMS/MS	[95]
103	MC-OiaAbu ^b^	C_52_H_68_N_8_O_13_	1012.4906	LC-MS/MS, thiol	[90]
104	MC-MR	C_48_H_72_N_10_O_12_S	1012.5052	LC-MS/MS, thiol, *S*-oxidation	[93]
105	[Mser^7^]MC-LR	C_49_H_76_N_10_O_13_	1012.5593	LC-HRMS	[80,94]
106	[seco-4/5]MC-LR ^b^	C_49_H_76_N_10_O_13_	1012.5593	LC-MS/MS, HRMS, thiol, NMR	[93,154]
107	[seco-1/2]MC-LR ^b^	C_49_H_76_N_10_O_13_	1012.5593	MS/MS, HRMS, NMR	[154]
108	MC-NfkA ^b^	C_51_H_66_N_8_O_14_	1014.4698	NMR, LC-MS/MS, HRMS, thiol, semisynthesis	[90]
109	[D-Asp³]MC-M(O)R ^b^	C_47_H_70_N_10_O_13_S	1014.4845	LC-MS/MS, thiol, *S*-oxidation	[93]
110	[D-Asp³,Dha^7^]MC-HphR	C_51_H_70_N_10_O_12_	1014.5175	LC-MS/MS	[33]
111	[D-Asp^3^]MC-FR	C_51_H_70_N_10_O_12_	1014.5175	AA, MS, NMR	[87]
112	[Dha^7^]MC-FR	C_51_H_70_N_10_O_12_	1014.5175	AA, HRMS, MS/MS	[155]
113	[DMAdda^5^]MC-FR	C_51_H_70_N_10_O_12_	1014.5175	LC-MS/MS	[137]
114	[D-Asp³]MC-RF	C_51_H_70_N_10_O_12_	1014.5175	LC-MS/MS, thiol	[108]
115	[Ser^7^]MC-E(OMe)E(OMe) ^b^	C_48_H_69_N_7_O_17_	1015.4750	HRMS, MS/MS	[138]
116	MC-LHty	C_53_H_73_N_7_O_13_	1015.5266	LC-MS/MS	[75]
117	[D-Asp³,Dha^7^]MC-RY	C_50_H_68_N_10_O_13_	1016.4967	LC-MS/MS, thiol	[108]
118	[D-Asp³,DMAdda^5^]MC-RY	C_50_H_68_N_10_O_13_	1016.4967	LC-MS/MS, thiol	[109]
119	MC-YM	C_51_H_69_N_7_O_13_S	1019.4674	AA, NMR, MS	[16]
120	[D-Asp^3^,ADMAdda^5^]MC-LHar	C_50_H_74_N_10_O_13_	1022.5437	HRMS, MS/MS, AA	[86]
121	[ADMAdda^5^]MC-LR	C_50_H_74_N_10_O_13_	1022.5437	HRMSNMR, AA, MS/MS	[78,79]
122	[D-Leu^1^,DMAdda^5^]MC-LR	C_51_H_78_N_10_O_12_	1022.5801	LC-HRMS/MS, thiol	[81]
123	[D-Leu^1^,dmAdda^5^]MC-LR (isomer 1) ^c^	C_51_H_78_N_10_O_12_	1022.5801	LC-HRMS/MS, thiol	[81]
124	[D-Leu^1^,dmAdda^5^]MC-LR (isomer 2) ^c^	C_51_H_78_N_10_O_12_	1022.5801	LC-HRMS/MS, thiol	[81]
125	[D-Leu^1^,D-Asp^3^]MC-LR	C_51_H_78_N_10_O_12_	1022.5801	LC-MS/MS, HRMS/MS	[152,153]
126	[D-Leu^1^,Dha^7^]MC-LR	C_51_H_78_N_10_O_12_	1022.5801	LC-MS/MS	[75]
127	[D-Val^1^]MC-LR	C_51_H_78_N_10_O_12_	1022.5801	LC-MS/MS	[75]
128	[Gly^1^,D-Asp³,Dhb^7^]MC-RHar	C_48_H_73_N_13_O_12_	1023.5502	LC-MS/MS, HRMS, AA, thiol	[91]
129	[D-Asp^3^]MC-RR	C_48_H_73_N_13_O_12_	1023.5502	AA, HRMS, NMR	[76,156]
130	[Dha^7^]MC-RR	C_48_H_73_N_13_O_12_	1023.5502	AA, HRMS, MS/MS, NMR	[157]
131	[D-Asp^3^,(*E*)-Dhb^7^]MC-RR	C_48_H_73_N_13_O_12_	1023.5502	NMR, HRMS	[158]
132	[Gly^1^,D-Asp^3^]MC-RHar	C_48_H_73_N_13_O_12_	1023.5502	LC-MS/MS	[92]
133	[DMAdda^5^]MC-RR	C_48_H_73_N_13_O_12_	1023.5502	LC-HRMS/MS, thiol	[82]
134	MC-WL	C_54_H_72_N_8_O_12_	1024.5270	LC-MS/MS, thiol	[90]
135	MC-LW	C_54_H_72_N_8_O_12_	1024.5270	LC-MS/MS, ^15^N-enrichment	[140]
136	[D-Asp^3^]MC-RCit	C_48_H_72_N_12_O_13_	1024.5342	LC-HRMS/MS, thiol, ^15^N-label	[99]
137	[D-Asp^3^,ADMAdda^5^,Thr^7^]MC-LR	C_49_H_74_N_10_O_14_	1026.5386	LC-MS/MS, thiol	[159]
138	[Seco-1/2]MC-HilR ^b^	C_50_H_78_N_10_O_13_	1026.5750	MS/MS, HRMS	[80]
139	[D-Asp^3^,Ser^7^]MC-RR	C_47_H_73_N_13_O_13_	1027.5451	LC-MS/MS, thiol	[159]
140	MC-NfkAbu ^b^	C_52_H_68_N_8_O_14_	1028.4855	LC-MS/MS, thiol	[90]
141	MC-M(O)R ^b^	C_48_H_72_N_10_O_13_S	1028.5001	AA, HRMS, NMR	[76]
142	MC-FR	C_52_H_72_N_10_O_12_	1028.5331	AA, HRMS, NMR	[76,160]
143	MC-RF	C_52_H_72_N_10_O_12_	1028.5331	LC-MS/MS, thiol	[77]
144	[D-Asp³]MC-HphR	C_52_H_72_N_10_O_12_	1028.5331	MS/MS	[88]
145	[Dha^7^]MC-HphR	C_52_H_72_N_10_O_12_	1028.5331	AA, HRMS, MS/MS, ^1^H-NMR	[161]
146	[D-Asp³]MC-M(O2)R ^b^	C_47_H_70_N_10_O_14_S	1030.4794	LC-MS/MS, thiol, *S*-oxidation	[93]
147	[D-Asp^3^,Dha^7^]MC-HtyR	C_51_H_70_N_10_O_13_	1030.5124	AA, HRMS, MS/MS, NMR	[161]
148	[D-Asp³,DMAdda^5^]MC-HtyR	C_51_H_70_N_10_O_13_	1030.5124	MS/MS	[88]
149	[Dha^7^]MC-YR	C_51_H_70_N_10_O_13_	1030.5124	HRMS, MS/MS, AA	[162]
150	[D-Asp^3^]MC-RY	C_51_H_70_N_10_O_13_	1030.5124	HRMS, LC-MS/MS, thiol	[108,163,164]
151	[Dha^7^]MC-RY	C_51_H_70_N_10_O_13_	1030.5124	LC-MS/MS, thiol	[77]
152	[D-Asp³,Dhb^7^]MC-RY	C_51_H_70_N_10_O_13_	1030.5124	LC-MS/MS, thiol	[109]
153	[D-Asp^3^]MC-YR	C_51_H_70_N_10_O_13_	1030.5124	AA, HRMS, MS/MS	[165]
154	[D-Asp^3^,(*E*)-Dhb^7^]MC-YR	C_51_H_70_N_10_O_13_	1030.5124	NMR, LC-HRMS/MS	[95]
155	[DMAdda^5^]MC-YR	C_51_H_70_N_10_O_13_	1030.5124	LC–MS/MS, thiol	[131]
156	MC-LY(OMe)	C_53_H_73_N_7_O_14_	1031.5216	LC-MS/MS, thiol	[108]
157	[D-Asp³]MC-(H4)YR	C_51_H_74_N_10_O_13_	1034.5437	LC–MS/MS, thiol	[131]
158	[Dha^7^]MC-(H4)YR	C_51_H_74_N_10_O_13_	1034.5437	HRMS, NMR, AA	[145]
159	[DMAdda^5^]MC-(H4)YR	C_51_H_74_N_10_O_13_	1034.5437	LC-MS/MS	[137]
160	MC-YM(O) ^b^	C_51_H_69_N_7_O_14_S	1035.4623	AA, NMR, MS	[166]
161	[ADMAdda^5^]MC-HilR	C_51_H_76_N_10_O_13_	1036.5593	LC-MS/MS	[33]
162	[ADMAdda^5^]MC-LHar	C_51_H_76_N_10_O_13_	1036.5593	HRMS, NMR, AA, MS/MS	[78,79]
163	MC-AnaR	C_52_H_80_N_10_O_12_	1036.5957	LC–MS/MS, thiol	[131]
164	[D-Leu^1^]MC-LR	C_52_H_80_N_10_O_12_	1036.5957	NMR, HRMS, MS/MS, AA	[167,168]
165	[D-Asp³]MC-YY	C_54_H_67_N_7_O_14_	1037.4746	LC-MS/MS, thiol	[108]
166	[Gly^1^,D-Asp^3^,ADMAdda^5^,Dhb^7^]MC-RR	C_48_H_71_N_13_O_13_	1037.5294	LC-MS/MS, HRMS, AA, thiol	[91]
167	[Gly^1^,D-Asp^3^,ADMAdda^5^]MC-RR	C_49_H_71_N_13_O_13_	1037.5294	LC-MS/MS	[92]
168	MC-RR	C_49_H_75_N_13_O_12_	1037.5658	NMR, AA, MS	[48]
169	[(6*Z*)-Adda^5^]MC-RR ^b^	C_49_H_75_N_13_O_12_	1037.5658	NMR, AA, MS	[115,147]
170	[D-Asp^3^,D-Glu(OMe)^6^]MC-RR ^b^	C_49_H_75_N_13_O_12_	1037.5658	NMR, AA, MS/MS	[169]
171	[D-Asp^3^]MC-RHar	C_49_H_75_N_13_O_12_	1037.5658	LC-HRMS/MS	[170]
172	[D-Ser^1^,ADMAdda^5^]MC-LR	C_50_H_74_N_10_O_14_	1038.5386	HRMS, MS/MS, AA	[86]
173	[D-Met^1^,D-Asp^3^]MC-LR	C_50_H_76_N_10_O_12_S	1040.5365	LC-MS/MS	[75]
174	[ADMAdda^5^,Mser^7^]MC-LR	C_50_H_76_N_10_O_14_	1040.5542	HRMS, MS/MS, AA	[86]
175	[Ser^7^]MC-RR	C_48_H_75_N_13_O_13_	1041.5607	AA, HRMS, MS/MS	[84]
176	[D-Asp³,Mser^7^]MC-RR	C_48_H_75_N_13_O_13_	1041.5607	AA, HRMS, MS/MS	[114]
177	[D-Asp³,Thr^7^]MC-RR	C_48_H_75_N_13_O_13_	1041.5607	MS/MS	[171]
178	[seco-1/6][D-Asp^3^]MC-RR ^b^	C_48_H_75_N_13_O_13_	1041.5607	NMR, AA, MS/MS	[169]
179	MC-HphR	C_53_H_74_N_10_O_12_	1042.5488	LC-MS/MS, thiol	[34,93]
180	[D-Glu(OMe)^6^]MC-FR ^b^	C_53_H_74_N_10_O_12_	1042.5488	LC-MS/MS	[40]
181	[D-Leu^1^]MC-LY	C_55_H_77_N_7_O_13_	1043.5579	LC-HRMS/MS, LC-MS/MS	[97,172]
182	MC-M(O_2_)R ^b^	C_48_H_72_N_10_O_14_S	1044.4950	LC-MS/MS, thiol, *S*-oxidation	[93]
183	[D-Asp^3^]MC-HtyR	C_52_H_72_N_10_O_13_	1044.5280	AA, MS, NMR	[173]
184	[Dha^7^]MC-HtyR	C_52_H_72_N_10_O_13_	1044.5280	AA, HRMS, MS/MS, ^1^H-NMR	[161]
185	[D-Asp^3^,(*E*)-Dhb^7^]MC-HtyR	C_52_H_72_N_10_O_13_	1044.5280	NMR, AA, HRMS	[142]
186	[D-Asp^3^,(*Z*)-Dhb^7^]MC-HtyR	C_52_H_72_N_10_O_13_	1044.5280	NMR, AA, HRMS	[142]
187	MC-RY	C_52_H_72_N_10_O_13_	1044.5280	LC-MS/MS, NMR, thiol	[77,163]
188	MC-YR	C_52_H_72_N_10_O_13_	1044.5280	AA, NMR, MS	[16]
189	[Dhb^7^]MC-YR	C_52_H_72_N_10_O_13_	1044.5280	LC-HRMS/MS, thiol	[174]
190	[seco-1/2]MC-FR ^b^	C_52_H_74_N_10_O_13_	1046.5437	HRMS, MS/MS	[80,154]
191	MC-(H2)YR	C_52_H_74_N_10_O_13_	1046.5437	LC-HRMS/MS, thiol	[82]
192	MC-HphHph	C_57_H_73_N_7_O_12_	1047.5317	LC-MS/MS	[75]
193	[D-Asp³,Ser^7^]MC-HtyR	C_51_H_72_N_10_O_14_	1048.5229	LC-MS/MS	[33]
194	[D-Asp³,Mser^7^]MC-RY	C_51_H_72_N_10_O_14_	1048.5229	LC-MS/MS, thiol	[108]
195	[Ser^7^]MC-YR	C_51_H_72_N_10_O_14_	1048.5229	LC–MS/MS	[102]
196	MC-(H4)YR	C_52_H_76_N_10_O_13_	1048.5593	HRMS, MS/MS, NMR	[80]
197	[ADMAdda^5^]MC-HilHar	C_52_H_78_N_10_O_13_	1050.5750	LC-MS/MS	[33]
198	[D-Leu^1^, D-Asp³,ADMAdda^5^]MC-LR	C_52_H_78_N_10_O_13_	1050.5750	LC-MS/MS	[153]
199	[D-Leu^1^,Adda(O)^5^]MC-LR ^b^	C_52_H_78_N_10_O_13_	1050.5750	LC-MS/MS, thiol	[81]
200	[D-Leu^1^]MC-HilR	C_53_H_82_N_10_O_12_	1050.6114	LC-MS/MS	[75]
201	[D-Leu^1^]MC-LHar	C_53_H_82_N_10_O_12_	1050.6114	LC-MS/MS	[75]
202	[D-Leu^1^,Glu(OMe)^6^]MC-LR ^b^	C_53_H_82_N_10_O_12_	1050.6114	HRMS/MS	[152]
203	[Hil^1^]MC-LR	C_53_H_82_N_10_O_12_	1050.6114	LC-MS/MS, thiol	[81]
204	[D-Asp³,(*E*)-Dhb^7^]MC-HtyY	C_55_H_69_N_7_O_14_	1051.4902	AA, NMR, HRMS	[175]
205	MC-YY	C_55_H_69_N_7_O_14_	1051.4902	NMR, LC-HRMS/MS	[95]
206	[Gly^1^,D-Asp^3^,ADMAdda^5^]MC-RHar	C_49_H_73_N_13_O_13_	1051.5451	LC-MS/MS	[92]
207	[Gly^1^,D-Asp^3^,ADMAdda^5^,Dhb^7^]MC-RHar	C_49_H_73_N_13_O_13_	1051.5451	LC-MS/MS, HRMS, AA, thiol	[91]
208	[D-Asp³,ADMAdda^5^]MC-RR	C_49_H_73_N_13_O_13_	1051.5451	LC-MS/MS	[153]
209	[D-Asp^3^,ADMAdda^5^,Dhb^7^]MC-RR	C_49_H_73_N_13_O_13_	1051.5451	NMR, HRMS, AA, LC-MS/MS, thiol	[150,176]
210	[D-Glu(OC_3_H_6_O)^6^]MC-LR ^b^	C_52_H_80_N_10_O_13_	1052.5906	AA, HRMS, NMR	[76]
211	[D-Leu^1^,Adda(OH)^5^]MC-LR ^b^	C_52_H_80_N_10_O_13_	1052.5906	LC-MS/MS, thiol	[81]
212	[D-Asp^3^]MC-WR	C_53_H_71_N_11_O_12_	1053.5284	AA, NMR, MS	[87]
213	[Dha^7^]MC-WR	C_53_H_71_N_11_O_12_	1053.5284	LC–MS/MS, thiol	[131]
214	[DMAdda^5^]MC-WR	C_53_H_71_N_11_O_12_	1053.5284	LC–MS/MS, thiol	[131]
215	[D-Asp^3^]MC-RW	C_53_H_71_N_11_O_12_	1053.5284	LC-HRMS/MS, thiol, ^15^N-label	[99]
216	[D-Met^1^]MC-LR	C_51_H_78_N_10_O_12_S	1054.5521	LC-MS/MS	[75]
217	[D-Leu^1^]MC-MR	C_51_H_78_N_10_O_12_S	1054.5521	LC-HRMS/MS, thiol, *S*-oxidation	[81]
218	[D-Leu^1^,Mser^7^]MC-LR	C_52_H_82_N_10_O_13_	1054.6063	HRMS/MS	[152]
219	[Mser^7^]MC-RR	C_49_H_77_N_13_O_13_	1055.5764	LC-HRMS/MS, thiol	[82]
220	[ADMAdda^5^]MC-FR	C_53_H_72_N_10_O_13_	1056.5280	LC-MS/MS	[137]
221	[D-Asp³,ADMAdda^5^]MC-HphR	C_53_H_72_N_10_O_13_	1056.5280	LC-MS/MS	[153]
222	MC-HtyR	C_53_H_74_N_10_O_13_	1058.5437	AA, MS, NMR	[173]
223	[D-Asp³,D-Glu(OMe)^6^]MC-HtyR ^b^	C_53_H_74_N_10_O_13_	1058.5437	MS/MS	[88]
224	[D-Glu(OMe)^6^]MC-YR ^b^	C_53_H_74_N_10_O_13_	1058.5437	NMR, LC-HRMS/MS	[95]
225	[D-Ser^1^,D-Asp³]MC-HtyR	C_52_H_72_N_10_O_14_	1060.5229	LC-MS/MS, thiol	[93]
226	[D-Asp³]MC-Y(OMe)R	C_52_H_72_N_10_O_14_	1060.5229	LC–MS/MS, thiol	[131]
227	[DMAdda^5^]MC-Y(OMe)R	C_52_H_72_N_10_O_14_	1060.5229	LC–MS/MS, thiol	[131]
228	[seco-4/5][D-Asp³]MC-HtyR ^b^	C_52_H_74_N_10_O_14_	1062.5386	LC-MS/MS, thiol	[93]
229	[Ser^7^]MC-HtyR	C_52_H_74_N_10_O_14_	1062.5386	AA, HRMS, MS/MS, NMR	[161]
230	[D-Asp³,Mser^7^]MC-HtyR	C_52_H_74_N_10_O_14_	1062.5386	MALDI-TOF MS	[88]
231	[D-Asp³,ADMAdda^5^]MC-(H4)YR	C_52_H_74_N_10_O_14_	1062.5386	LC-MS/MS	[153]
232	[Mser^7^]MC-RY	C_52_H_74_N_10_O_14_	1062.5386	LC-MS/MS, thiol	[77]
233	[D-Asp³,Mser^7^]MC-YHar	C_52_H_74_N_10_O_14_	1062.5386	NMR, LC-HRMS/MS	[95]
234	[Mser^7^]MC-YR	C_52_H_74_N_10_O_14_	1062.5386	LC–MS/MS, NMR, thiol, HRMS	[131]
235	MC-HphHty	C_57_H_73_N_7_O_13_	1063.5266	LC-MS/MS	[75]
236	[D-Leu^1^,ADMAdda^5^]MC-LR	C_53_H_80_N_10_O_13_	1064.5906	LC –MS/MS	[153]
237	[D-Leu^1^,Glu(OMe)^6^]MC-HilR ^b^	C_54_H_84_N_10_O_12_	1064.6270	HRMS/MS	[152]
238	[D-Asp³,(*E*)-Dhb^7^]MC-HtyHty	C_56_H_71_N_7_O_14_	1065.5059	AA, NMR, HRMS	[175]
239	[ADMAdda^5^]MC-RR	C_50_H_75_N_13_O_13_	1065.5607	LC-MS/MS, LC-HRMS/MS	[104,153]
240	MC-HarHar	C_51_H_79_N_13_O_12_	1065.5971	LC–MS/MS	[177]
241	[D-Leu^1^,D-Asp^3^]MC-RR	C_51_H_79_N_13_O_12_	1065.5971	LC-MS/MS	[75]
242	MC-WR	C_54_H_73_N_11_O_12_	1067.5440	AA, HRMS, NMR	[76]
243	[D-Leu^1^,D-Glu(OMe)^6^,Mser^7^]MC-LR ^b^	C_53_H_84_N_10_O_13_	1068.6219	HRMS/MS	[152]
244	[D-Met(O)^1^]MC-LR ^b^	C_51_H_78_N_10_O_13_S	1070.5471	HRMS/MS	[152]
245	[D-Leu^1^]MC-M(O)R ^b^	C_51_H_78_N_10_O_13_S	1070.5471	LC-HRMS/MS, thiol, *S*-oxidation	[81,97]
246	[D-Leu^1^,D-Asp³]MC-HphR	C_55_H_78_N_10_O_12_	1070.5801	LC-MS/MS	[153]
247	[D-Phe^1^]MC-LR	C_55_H_78_N_10_O_12_	1070.5801	LC-MS/MS	[75]
248	[D-Leu^1^]MC-FR	C_55_H_78_N_10_O_12_	1070.5801	LC-MS/MS, thiol	[81]
249	MC-KynR ^b^	C_53_H_73_N_11_O_13_	1071.5389	LC-MS/MS, thiol	[90]
250	[D-Asp^3^,ADMAdda^5^]MC-HtyR	C_53_H_72_N_10_O_14_	1072.5229	AA, HRMS, MS/MS	[178]
251	[D-Asp^3^,ADMAdda^5^,Dhb^7^]MC-HtyR	C_53_H_72_N_10_O_14_	1072.5229	NMR, HRMS, AA	[150]
252	[ADMAdda^5^]MC-YR	C_53_H_72_N_10_O_14_	1072.5229	LC-MS/MS	[33]
253	MC-MhtyR	C_54_H_76_N_10_O_13_	1072.5593	LC-HRMS/MS, thiol, esterification	[96]
254	[D-Asp^3^,(*E*)-Dhb^7^]MC-HtyW	C_57_H_70_N_8_O_13_	1074.5062	NMR, LC-HRMS/MS	[95]
255	MC-RY(OMe)	C_53_H_74_N_10_O_14_	1074.5386	LC-MS/MS, thiol	[77]
256	[D-Asp³]MC-Hty(OMe)R	C_53_H_74_N_10_O_14_	1074.5386	LC-MS/MS, thiol	[93]
257	MC-Y(OMe)R	C_53_H_74_N_10_O_14_	1074.5386	LC–MS/MS, thiol	[131]
258	[seco-4/5]MC-HtyR ^b^	C_53_H_76_N_10_O_14_	1076.5542	LC-MS/MS, thiol	[93]
259	[ADMAdda^5^]MC-(H4)YR	C_53_H_76_N_10_O_14_	1076.5542	LC-MS/MS	[137]
260	[Mser^7^]MC-HtyR	C_53_H_76_N_10_O_14_	1076.5542	LC-MS/MS, thiol	[93]
261	[D-Leu^1^,ADMAdda^5^]MC-LHar	C_54_H_82_N_10_O_13_	1078.6063	LC-MS/MS	[153]
262	[ADMAdda^5^]MC-RHar	C_51_H_77_N_13_O_13_	1079.5764	LC-MS/MS	[153]
263	[D-Leu^1^]MC-RR	C_52_H_81_N_13_O_12_	1079.6128	LC-MS/MS	[75]
264	MC-OiaR ^b^	C_54_H_73_N_11_O_13_	1083.5389	LC-MS/MS, thiol	[90]
265	[D-Met(O)^1^,Glu(OMe)^6^]MC-LR ^b^	C_52_H_80_N_10_O_13_S	1084.5627	HRMS/MS	[152]
266	[D-Leu^1^]MC-HphR	C_56_H_80_N_10_O_12_	1084.5957	LC-MS/MS	[75]
267	[D-Leu^1^]MC-M(O2)R ^b^	C_51_H_78_N_10_O_14_S	1086.5420	LC-HRMS/MS, thiol, *S*-oxidation	[81]
268	[ADMAdda^5^]MC-(H_4_)YHar	C_54_H_78_N_10_O_14_	1090.5699	LC-MS/MS	[137]
269	MC-NfkR ^b^	C_54_H_73_N_11_O_14_	1099.5338	LC-MS/MS, thiol	[90]
270	[D-Leu^1^]MC-HtyR	C_56_H_80_N_10_O_13_	1100.5906	LC-MS/MS	[153]
271	[D-Leu^1^,Ser^7^]MC-HtyR	C_55_H_80_N_10_O_14_	1104.5855	LC-HRMS/MS	[170]
272	MC-LR Cys conjugate ^b^	C_52_H_81_N_11_O_14_S	1115.5685	HRMS, MS/MS, semisynthesis	[80]
273	[D-Leu^1^]MC-LR Cys conjugate ^b^	C_55_H_87_N_11_O_14_S	1157.6155	LC-HRMS/MS, semisynthesis	[81]
274	[D-Leu^1^]MC-LR Cys sulfoxide conjugate ^b^	C_55_H_87_N_11_O_15_S	1173.6104	LC-HRMS/MS, semisynthesis, *S*-oxidation	[81]
275	[D-Leu^1^]MC-LR γ-GluCys conjugate ^b^	C_60_H_94_N_12_O_17_S	1286.6581	LC-HRMS/MS, semisynthesis	[81]
276	[D-Leu^1^]MC-LR γ-GluCys sulfoxide conjugate ^b^	C_60_H_94_N_12_O_18_S	1302.6530	LC-HRMS/MS, semisynthesis, *S*-oxidation	[81]
277	[D-Asp³]MC-RRGSH conjugate ^b^	C_58_H_90_N_16_O_18_S	1330.6340	LC-HRMS/MS, thiol, ^15^N-label, semisynthesis	[99]
278	[D-Leu^1^]MC-LR GSH conjugate ^b^	C_62_H_97_N_13_O_18_S	1343.6795	LC-HRMS/MS, semisynthesis	[81]
279	[D-Leu^1^]MC-LR GSH sulfoxide conjugate ^b^	C_62_H_97_N_13_O_19_S	1359.6744	LC-HRMS/MS, semisynthesis, *S*-oxidation	[81]

^a^ Exact monoisotopic neutral mass. For protonated/deprotonated ions, *m*/*z* = (monoisotopic neutral mass +/− (*z* × 1.0073))/*z*. ^b^ Congeners expected to form through chemical or biochemical transformations of other MCs rather than through biosynthesis by cyanobacteria, see Section 5.7 for additional details. ^c^ Entries 123 and 124 are a pair of congeners demethylated at different positions somewhere between C-2 and C-8 of their Adda moieties. *Characterization Techniques*. AA, amino acid analysis; MS, mass spectrometry (low resolution MS analysis of isolated MCs implied); NMR, nuclear magnetic resonance(including ^1^H, ^13^C or 2-dimensional experiments); MS/MS, tandem mass spectrometry (direct analysis of isolated fractions using either ESI or MALDI ionization); HRMS, high resolution mass spectrometry (direct analysis of isolated fractions); LC-MS/MS, liquid chromatography-tandem mass spectrometry (without isolation); LC-HRMS/MS, liquid chromatography-high resolution tandem mass spectrometry (without isolation); thiol, reactivity with a thiol-containing reagent (e.g., mercaptoethanol) to confirm presence of an α,β-unsaturated amide such as in Mdha^7^; *S*-oxidation, selective oxidation with mild oxidant (e.g., periodate) to confirm presence of methionine residues and thiol conjugates as well as their corresponding sulfones and sulfoxides; semisynthesis, chemical conversion of a known congener to the unknown one (e.g., by derivatization with GSH, or oxidation of Trp residues); esterification, derivatization of carboxylic acid groups as esters or phenolic groups as ethers; ^15^N-labelling, culturing in ^15^N-labelled medium to count N-atoms.

**Table 2 toxins-11-00714-t002:** Lethal dose (LD_50_) values (µg kg^-1^ b.w., i.p. mouse) of MC congeners.

Entry ^a^	Microcystin	LD_50_	Reference
7	MC-LA	50	[226]
18	MC-AR	250	[76]
22	MC-YA	60–70	[227]
25	[D-Asp^3^,Dha^7^]MC-LR	160–300	[83]
39	[D-Asp^3^]MC-LR	200–500	[139,215]
40	[D-Asp^3^,(*E*)-Dhb^7^]MC-LR	70	[142]
42	[Dha^7^]MC-LR	250	[83]
43	[DMAdda^5^]MC-LR	90–100	[76,149]
66	MC-LR	50	[205]
68	[(6*Z*)-Adda^5^]MC-LR	>1200	[115,147]
83	MC-LY	90	[206]
89	[D-Asp^3^,ADMAdda^5^]MC-LR	160	[78,79]
92	MC-HilR	100	[80]
94	[D-Glu(OMe)^6^]MC-LR	>1000	[85,140,189]
105	[Mser^7^]MC-LR	150	[80]
111	[D-Asp^3^]MC-FR	90 ± 10	[216]
121	[ADMAdda^5^]MC-LR	60	[78,79]
129	[D-Asp^3^]MC-RR	350 ± 10	[216]
130	[Dha^7^]MC-RR	180	[157]
131	[D-Asp^3^,(*E*)-Dhb^7^]MC-RR	250	[142]
141	MC-M(O)R	700–800	[76]
142	MC-FR	250	[76]
160	MC-YM(O)	56	[166]
162	[ADMAdda^5^]MC-LHar	60	[78,79]
164	[D-Leu^1^]MC-LR	100	[167,168]
168	MC-RR	500–800	[48,76]
169	[(6*Z*)-Adda^5^]MC-RR	>1200	[115,147]
183	[D-Asp^3^]MC-HtyR	80–100	[147]
185	[D-Asp^3^,(*E*)-Dhb^7^]MC-HtyR	70	[142]
188	MC-YR	70	[16]
209	[D-Asp^3^,ADMAdda^5^,Dhb^7^]MC-RR	200	[150]
210	[D-Glu(OC_3_H_6_O)^6^]MC-LR	>1000	[76]
212	[D-Asp^3^]MC-WR	95 ± 10	[216]
222	MC-HtyR	160–300	[147]
242	MC-WR	150–200	[76]
251	[D-Asp^3^, ADMAdda^5^,Dhb^7^]MC-HtyR	100	[150]
272	MC-LR Cys conjugate	1000	[80]

**^a^** Entry number of each MC as indicated in Table 1 and Table 3.

**Table 3 toxins-11-00714-t003:** The half inhibitory concentration (IC_50_) values (nM) for inhibition of serine/threonine protein phosphatases (PPs) by MC congeners.

Entry ^a^	Microcystin	Type of PP	Origin of PP	IC_50_	Reference
7	MC-LA	PP1	Rabbit muscle	2.3	[237]
		PP2A	Human hepatocytes	0.56	[238]
		PP2A	Rabbit muscle	0.05	[237]
		rPP2Ac	Recombinant human PP2A catalytic subunit	0.161 ± 0.002	[239]
14	MC-LV	PP1	Rabbit skeletal muscle	0.06–0.45	[133]
17	MC-LL	PP1	Rabbit skeletal muscle	0.06–0.45	[133]
25	[D-Asp^3^,Dha^7^] MC-LR	rPP2Ac	Recombinant human PP2A catalytic subunit	0.254 ± 0.004	[239]
34	MC-LM	PP1	Rabbit skeletal muscle	0.06–0.45	[133]
39	[D-Asp^3^]MC-LR	PP2A	Rabbit skeletal muscle	0.09	[215]
40	[D-Asp^3^,(*E*)-Dhb^7^]MC-LR	rPP2Ac	Recombinant human PP2A catalytic subunit	0.201 ± 0.003	[239]
41	[D-Asp^3^,(*Z*)-Dhb^7^]MC-LR	rPP2Ac	Recombinant human PP2A catalytic subunit	0.16 ± 0.01	[239]
42	[Dha^7^]MC-LR	PP1	Rabbit skeletal muscle	0.54–5	[240,241]
		PP2A	Bovine kidney	0.11 ± 0.04	[241]
		rPP2Ac	Recombinant human PP2A catalytic subunit	0.167 ± 0.003	[239]
43	[DMAdda^5^]MC-LR	rPP1c	Recombinant rabbit skeletal muscle PP1	1.5	[240]
52	MC-LF	rPP1c	Recombinant rabbit skeletal muscle PP1	1.8	[224]
		PP1	Rabbit skeletal muscle	0.06–0.45	[133]
		PP2A	Human hepatocytes	0.57	[238]
		PP2A	Human red blood cells	1.1	[224]
		rPP2Ac	Recombinant human PP2A catalytic subunit	0.10 ± 0.02	[239]
66	MC-LR	PP1	Rabbit muscle	0.1–1.9	[215,217,237,240,241,242,243,244,245]
		PP1	Chicken gizzard myosin B	6	[246]
		PP1	Liver of grass carp	0.90	[247]
		rPP1c	Recombinant rabbit skeletal muscle PP1	1.2	[224]
		PP2A	Rabbit skeletal muscle	0.04–0.5	[215,217,237,243,248]
		PP2A	Human hepatocytes	0.46	[238]
		PP2A	Human erythrocytes	0.03–2.2	[241,242,249,250]
		PP2A	Bovine heart	0.05-2	[241,246,251]
		PP2A	Bovine kidney	0.2 ± 0.1	[241]
		PP2A	Mouse brain	0.28–3.15	[218,247,252,253]
		PP2A	Liver of grass carp	0.28	[247]
		rPP2Ac	Recombinant human PP2A catalytic subunit	0.032 ± 0.004	[239]
68	[(6*Z*)-Adda^5^]MC-LR	PP2A	Mouse brain	80	[252]
		rPP1c	Recombinant rabbit skeletal muscle PP1	>100	[240]
83	MC-LY	PP2A	Human hepatocytes	0.34	[238]
89	[D-Asp^3^,ADMAdda^5^]MC-LR	PP2A^b^		4	[178]
94	[D-Glu(OMe)^6^]MC-LR	rPP1c	Recombinant rabbit skeletal muscle PP1	>100	[240]
97	[D-Asp^3^,Dha^7^]MC-RR	rPP1c	Recombinant rabbit skeletal muscle PP1	0.22 ± 0.01	[239]
129	[D-Asp^3^]MC-RR	PP2A rPP2Ac	Human red blood cells Recombinant human PP2A catalytic subunit	0.45–11.5 0.30 ± 0.01	[254] [239]
130	[Dha^7^]MC-RR	PP1	Rabbit muscle	8.3 ± 0.8	[241]
		PP2A	Bovine kidney	4 ± 1	[241]
		rPP2Ac	Recombinant human PP2A catalytic subunit	0.29 ± 0.01	[239]
		PP1	Rabbit skeletal muscle	2.6–5.7	[254]
131	[D-Asp^3^,(*E*)-Dhb^7^]MC-RR	PP1	Rabbit skeletal muscle	1.8–56.4	[215,254]
		PP2A	Rabbit skeletal muscle	2.4	[215]
		PP2A	Human red blood cells	17.9–49.4	[254]
135	MC-LW	rPP1c	Recombinant rabbit skeletal muscle PP1	1.9	[224]
		PP2A	Human hepatocytes	0.29	[238]
		PP2A	Human red blood cells	1.1	[224]
		rPP2Ac	Recombinant human PP2A catalytic subunit	0.114 ± 0.003	[239]
142	MC-FR	rPP2Ac	Recombinant human PP2A catalytic subunit	0.069 ± 0.003	[239]
149	[Dha^7^]MC-YR	rPP2Ac	Recombinant human PP2A catalytic subunit	0.379 ± 0.003	[239]
164	[D-Leu^1^]MC-LR	PP1	Recombinant rabbit skeletal muscle PP1	0.5–4.43	[167,168]
168	MC-RR	PP1	Rabbit skeletal muscle	0.68	[215]
		PP1	Chicken gizzard myosin B	3	[246]
		PP1	Liver of grass carp	3.60	[247]
		rPP1c	Recombinant rabbit skeletal muscle PP1	1.5	[224]
		PP2A	Human red blood cells	0.241–175	[241,249,250]
		PP2A	Human hepatocytes	0.60	[238]
		PP2A	Mouse brain	0.72–1.4	[218,224,247,252]
		PP2A	Bovine cardiac muscle	1	[246]
		PP2A	Bovine Kidney	10 ± 2	[241]
		PP2A	Rabbit skeletal muscle	0.1	[215]
		PP2A	Liver of grass carp	0.64	[247]
		rPP2Ac	Recombinant human PP2A catalytic subunit	0.056 ± 0.002	[239]
169	[(6*Z*)-Adda^5^]MC-RR	rPP2Ac	Recombinant human PP2A catalytic subunit	10.1 ± 0.3	[239]
		PP2A	Mouse brain	80	[252]
183	[D-Asp^3^]MC-HtyR	rPP2Ac	Recombinant human PP2A catalytic subunit	0.098 ± 0.006	[239]
185	[D-Asp^3^,(*E*)-Dhb^7^]MC-HtyR	rPP2Ac	Recombinant human PP2A catalytic subunit	0.122 ± 0.005	[239]
186	[D-Asp^3^,(*Z*)-Dhb^7^]MC-HtyR	rPP2Ac	Recombinant human PP2A catalytic subunit	0.110 ± 0.008	[239]
188	MC-YR	PP1 PP1 PP2A PP2A PP2A PP2A PP2A PP2A rPP2Ac	Rabbit skeletal muscle Liver of grass carp Human red blood cells Human hepatocytes Bovine kidney Rabbit skeletal muscle Mouse brain Liver of grass carp Recombinant human PP2A catalytic subunit	1.0 0.90 0.26–9.0 0.84 0.09 ± 0.02 0.26 0.39–1.3 0.40 0.125 ± 0.005	[215] [247] [241,249,250] [238] [241] [215] [218,247] [247] [239]
242	MC-WR	rPP2Ac	Recombinant human PP2A catalytic subunit	0.18 ± 0.01	[239]
251	[D-Asp^3^,ADMAdda^5^,Dhb^7^]MC-HtyR	PP1	Rabbit skeletal muscle	0.15–0.24	[254]
		PP2A	Human red blood cells	0.06	[254]
		PP2A	Bovine heart	0.06	[251]
		PP4	Porcine testis	0.04	[251]
		PP5	Recombinant human PP5 expressed in *E. coli*	0.5	[251]

**^a^** Entry number of each MC as indicated in Table 1 and Table 2
^b^ Not specified.

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
