# Peer review of "Structural Diversity, Characterization and Toxicology of Microcystins"

_toxins, 2019, doi:10.3390/toxins11120714_

Round 1

Reviewer 1 Report

I really enjoyed reading this review.

1) Please make sure that you either always spell out microcystin, or define and use abbreviation throughout the text.

2) RPLC is not defined.

3) SPE is not defined.

4) All NMR - TOCSY, COSY all need to be defined.

5) 699 sentence is not complete.

Author Response

Many thanks for your glowing comment and corrections on our manuscript. We have used all of them (marked changes in red in the manuscript) to correct and rewrite the paper. The review has been updated with the addition of 10 new microcystin congeners that were recently published in October-November 2019.

Reviewer 1

Comments and Suggestions for Authors

I really enjoyed reading this review.

1) Please make sure that you either always spell out microcystin, or define and use abbreviation throughout the text.

Done

2) RPLC is not defined.

Done

3) SPE is not defined.

Done

4) All NMR - TOCSY, COSY all need to be defined.

Done

5) 699 sentence is not complete.

The sentence was corrected

Reviewer 2 Report

The authors are reporting a review study about the current state of knowledge on the diversity of microcystin congeners, its characterization and associated toxicology.

The manuscript congregates important information on a thematic, which is often scattered and incomplete in many papers. At the same time the authors have identified important gaps in knowledge on this issue. The manuscript is consistent; it is very well written and has innovative and suitable publishable data, which will certainly be of interest to the readers of Toxins.

However, some minor revisions will be necessary and some other suggestions could improve the manuscript.

Minor comments:

Abstract

L7: I would recommend replace ‘…structure of the first MCs…’ by ‘…structure of MC was firstly…’; L18 and 19: I would recommend writing MCs congeners/variants; L21: I would recommend replace the word ‘cyanobacterial’ by ‘cyanobacteria’; L27: Keywords –The keyword microcystin should be replaced by microcystin congeners.

Introduction: L38: I would recommend replace ‘…human organs on which thy act…’ by ‘…target organs…’; L43: I would recommend after human health remove the ‘,’; L44: The reference number 8 is regarding neurotoxins alkaloids (anatoxins/saxitoxins). Why are the authors referencing that if are talking about MCs? L47: I would recommend replace ‘He…’ by ‘With this study the authors…’ L55: The abbreviation MC was not defined for the first time.

Nomenclature …. L93: A reference should be added at the end of this sentence. L111: MCs was abbreviated for the first time in L40. Then, the same abbreviation should be used for the rest of the manuscript. This inconsistency is presented in other parts of manuscript (e.g., lines 113, 144; 164; 175; 199; 216; 226, etc.). Biosynthesis …. L128-129: Why the authors do not specify the nitrogen as an important environmental factor and then describes detailed information about that? The contrary for temperature and pH since it is just mentioned here. Please revise. L149: I would recommend replace ‘concentrations…’ by ‘concentration…’

Structural ….

- L176-178: This sentence is confusing. The writing should be improved.

- L187 and190: Liquid chromatography was defined for the first time in L184. Then, the same abbreviation should be used for the rest of the manuscript.

- L194: I would recommend replace ‘…never the less…’ by ‘…nevertheless…’;

- L207; 212; 229; 263; 301 and 310: A reference should be added at the end of these sentences.

- L293: SPE should be defined for the first time;

- L326: TOCSY should be defined for the first time;

- L328, 329; 333; 337 and 338: The abbreviations should be defined for the first time;

Diversity….

-L460: I would recommend replace ‘…it’s not…’ by ‘…it is not…’;

- L564: A reference should be added at the end of this sentence.

L582: GSH was defined for the first time in L570.

Toxicity…

- L628-629: It would be important to consider the potential effects of digestion on MCs bioavailability. There are some studies showing that the digestion of MCs with pepsin, tripsin and chymotrypsin can reduce the MC-LR bioaccessibility and consequently the bioavailability, potentially reducing the toxicity by oral ingestion.

- L689: A reference should be added at the end of this sentence.

Author Response

Reviewer 2

Comments and Suggestions for Authors

The authors are reporting a review study about the current state of knowledge on the diversity of microcystin congeners, its characterization and associated toxicology.

The manuscript congregates important information on a thematic, which is often scattered and incomplete in many papers. At the same time the authors have identified important gaps in knowledge on this issue. The manuscript is consistent; it is very well written and has innovative and suitable publishable data, which will certainly be of interest to the readers of Toxins.

However, some minor revisions will be necessary, and some other suggestions could improve the manuscript.

Many thanks for your very constructive comments and corrections on our manuscript. We have used all of them (marked changes in red in the manuscript) to correct and rewrite the paper. The review has been updated with the addition of 10 new microcystin congeners that were recently published in October-November 2019.

Minor comments:

Abstract

L7: I would recommend replace ‘…structure of the first MCs…’ by ‘…structure of MC was firstly…’;

The sentence as currently written is correct, and the grammar is correct. The reviewer’s suggestion is incorrect, because “microcystin” is not one compound, it is a class of compounds. Therefore, “microcystin” cannot have a structure to be elucidated. Therefore no change has been made.

L18 and 19: I would recommend writing MCs congeners/variants

Again, the grammar and meaning are clear and correct here. We have defined MCs an abbreviation for “microcystins”, and there would be nothing wrong with saying “at least 269 microcystins have been reported”. On the other hand, writing “microcystins variants” as suggested would be grammatically incorrect.

L21: I would recommend replace the word ‘cyanobacterial’ by ‘cyanobacteria’

Done

L27: Keywords –The keyword microcystin should be replaced by microcystin congeners.

Placing congeners as part of the search term makes it difficult for searchers. Should they search for “microcystin variants”, “microcystins”, or “microcystin congeners”? The key word here, common to all these potential search terms, is “microcystin”.

Introduction: L38: I would recommend replace ‘…human organs on which thy act…’ by ‘…target organs…’

Yes, good suggestion

L43: I would recommend after human health remove the ‘,’

Done

L44: The reference number 8 is regarding neurotoxins alkaloids (anatoxins/saxitoxins). Why are the authors referencing that if are talking about MCs?

In the same year (2016) the first author published two articles, one concerning the neurotoxins and a second concerning microcystins. So there was a confusion and the reference was well corrected in line with the text.

L47: I would recommend replace ‘He…’ by ‘With this study the authors…’

No, in fact there were two authors. So the citation in the previous sentence should be “Louw and Smit [13]…”, which has now been corrected, and consequently the next sentence should begin with either “They…” or “these authors…”.

L55: The abbreviation MC was not defined for the first time.

Done

Nomenclature …. L93: A reference should be added at the end of this sentence.

Done

L111: MCs was abbreviated for the first time in L40. Then, the same abbreviation should be used for the rest of the manuscript. This inconsistency is presented in other parts of manuscript (e.g., lines 113, 144; 164; 175; 199; 216; 226, etc.).

Yes, we have now changed this.

Biosynthesis …. L128-129: Why the authors do not specify the nitrogen as an important environmental factor and then describes detailed information about that? The contrary for temperature and pH since it is just mentioned here. Please revise.

A good point. Nutrient availability is now also listed as a factor.

L149: I would recommend replace ‘concentrations…’ by ‘concentration…’

Done

Structural ….

L176-178: This sentence is confusing. The writing should be improved.

The sentence was corrected

L187 and190: Liquid chromatography was defined for the first time in L184. Then, the same abbreviation should be used for the rest of the manuscript.

Done

- L194: I would recommend replace ‘…never the less…’ by ‘…nevertheless…’

Done

- L207; 212; 229; 263; 301 and 310: A reference should be added at the end of these sentences.

All references were added

L293: SPE should be defined for the first time;

Done

L326: TOCSY should be defined for the first time;

Done

L328, 329; 333; 337 and 338: The abbreviations should be defined for the first time;

Done

Diversity….

-L460: I would recommend replace ‘…it’s not…’ by ‘…it is not…’;

Done

- L564: A reference should be added at the end of this sentence.

Done

L582: GSH was defined for the first time in L570.

Done

Toxicity…

- L628-629: It would be important to consider the potential effects of digestion on MCs bioavailability. There are some studies showing that the digestion of MCs with pepsin, tripsin and chymotrypsin can reduce the MC-LR bioaccessibility and consequently the bioavailability, potentially reducing the toxicity by oral ingestion.

Done

- L689: A reference should be added at the end of this sentence.

Done

Reviewer 3 Report

This is a beautiful manuscript that clears the known information about cyanotoxins. 

There are few details (some punctuation details, mostly) that are important but easily improved.

I think it is almost ready for publication. Congratulations.

Author Response

Many thanks for your glowing comment. The review has been updated with the addition of 10 new microcystin congeners that were recently published in October-November 2019.

Reviewer 4 Report

In this review, the authors comprehensively address the diversity of microcystins and how to detect via MS methodologies. I think that the manuscript is well-written and when published it will be a useful resource for the community.

Author Response

(The authors gave the same response as above.)
